# HSP90 as an evolutionary capacitor drives adaptive eye size reduction via *atonal*

Rascha Sayed [1], Özge Şahin[1,3], Mohammed Errbii [1,3], Reshma R [1], Robert Peuß [2], Tobias Prüser[1], Lukas Schrader [1], Nora K. E. Schulz [1] & Joachim Kurtz [1]✉

Genetic variation fuels evolution, and the release of cryptic variation is key for adaptation. The heat shock protein 90 (HSP90) has been proposed to act as an evolutionary capacitor by revealing such hidden variation under stress. However, this idea remains debated, as the genetic basis of HSP90-regulated traits is often unknown, and many observed phenotypes are deleterious. Here, we show in *Tribolium castaneum* that HSP90 shapes evolution by unmasking a hidden trait providing enhanced fitness under specific conditions. Using RNA interference and chemical inhibition, we consistently reveal a *reduced-eye* phenotype that persists in descendant lines across generations without continued HSP90 disruption. Under constant light, *reduced-eye* beetles had higher reproductive success and greater trait penetrance than normal-eyed siblings, suggesting a selective advantage. Whole-genome sequencing and functional analysis identify the transcription factor *atonal* (*ato*) as the underlying gene. These results provide the first direct genetic link between an HSP90-buffered trait and context-dependent fitness benefits in animals, highlighting a potential mechanism by which cryptic variation contributes to adaptation.

Canalization stabilizes the phenotype against genetic and environmental perturbations[1,2]. Heat shock protein 90 (HSP90) contributes to this process by buffering against biochemical, genetic and epigenetic influences[3,4]. HSP90 is primarily a molecular chaperone that maintains protein homeostasis by assisting correct protein folding[5,6], directly or indirectly interacting with approximately 10% of the proteome[7]. Many client proteins of HSP90 are involved in the regulation of developmental pathways and genomic stability[8]. Consequently, impaired HSP90 function, for example in Drosophila *Hsp83* mutants, results in a multitude of morphological changes in wings, eyes, and legs[3]. As artificial selection has been shown to maintain these altered phenotypes in subsequent generations, HSP90 has been described as an evolutionary capacitor, referring to its ability to store and release genetic variation[3,9,10].

As long as HSP90 prevents genetic variants from being expressed phenotypically (*i.e.*, maintains genetic variation in a cryptic state),

selection will not act on such variants and phenotypic evolution will be temporarily constrained[11–13]. However, under stressful environmental conditions that disrupt proteostasis, HSP90 availability may become limited due to its involvement in stabilizing numerous proteins that are damaged or denatured during stress[14–17]. In these situations, stored genetic variation may be released and expressed as phenotypic differences on which selection can act either in the disrupting environmental conditions or other conditions altogether. Therefore, HSP90 serves as a molecular mechanism driving the de-canalization and potential subsequent assimilation of traits[3,9].

Additional functions of HSP90 have been described that might likewise support its role in facilitating rapid adaptation. HSP90 serves as a capacitor for epigenetic variation, regulating gene expression through chromatin remodeling[4,18], and silencing endogenous retroviruses' influence on the transcription of nearby genes[19]. A recent study in yeast demonstrated that HSP90 buffers *cis*-regulatory variation,

[1]Institute for Evolution and Biodiversity, University of Münster, Hüfferstrasse 1, Münster, Germany. [2]Institute for Integrative Cell Biology and Physiology, University of Münster, Schlossplatz 8, Münster, Germany. [3]These authors contributed equally: Özge Şahin, Mohammed Errbii. ✉e-mail: joachim.kurtz@uni-muenster.de

particularly related to dosage-sensitive genes[20]. HSP90 reduction has also been suggested to increase the activity of transposable elements (TEs), thereby generating new genetic variation[21].

The hypothesis that HSP90 acts as an evolutionary capacitor is based on the idea that some of the buffered phenotypes may facilitate adaptability in specific environments. However, the vast majority of phenotypes released by HSP90 inhibition in animals is detrimental[16,22,23]. While studies in yeast could demonstrate fitness benefits of HSP90-buffered traits[20,24,25], compelling evidence directly linking HSP90-buffered phenotypes to fitness advantages in animals is still largely absent. For example, in surface populations of cavefish *Astyanax mexicanus*, HSP90 inhibition induced variation in eye size, suggesting that the evolution of adaptive eye loss in caves might have been facilitated by HSP90[12]. However, the genetic basis of HSP90-regulated eye-size variation remained unknown. The only insect study on HSP90-regulated traits, conducted in *Drosophila melanogaster*, found the deformed eye trait—first identified by Rutherford and Lindquist (1998)—to be neutral, with no adaptive benefits in any tested environment[26]. Since most identified HSP90-buffered variants were deleterious, their contribution to adaptation stayed uncertain[16,22]. Furthermore, while several HSP90-dependent loci have been mapped in natural populations and specific systems[10,24], the genetic basis of HSP90-regulated phenotypes has largely remained elusive in animals[27], while studies in yeast allowed for large-scale linking to the genetic level[20,24]. To prove a role of HSP90 as an evolutionary capacitor, it is mandatory to uncover both, the genetic basis of HSP90-relesased phenotypes and their fitness relevance.

The red flour beetle, *Tribolium castaneum*, is an established alternative insect model for studying developmental and evolutionary genetics[28,29]. A previous study in *T. castaneum* has shown consistent downregulation of *Hsp83*, the primary HSP90-coding gene, in response to social cues mimicking a stressful environment, suggesting that HSP90 levels may be adaptively regulated and could facilitate the release of cryptic genetic variation when advantageous[17]. Here we directly manipulate HSP90 function in *T. castaneum*, characterize the resulting *reduced-eye* phenotype, identify the gene underlying this phenotype, and assess its fitness consequences to evaluate the adaptive potential of HSP90-released variation.

## Results

### HSP90 inhibition releases a *reduced-eye* phenotype
To investigate the effects of reduced HSP90 function—mimicking conditions that may arise during environmental stress—we used RNA interference (RNAi) to target *Hsp83* in *T. castaneum*, with the aim of releasing HSP90-buffered, selectable phenotypic variants in Cro1, a genetically heterogenous wildtype population of this beetle[30]. We hypothesized that such variants could confer fitness advantages under certain environmental conditions, leading to the fixation of novel adaptive traits from previously cryptic genetic variation if HSP90-dependent canalization is disrupted. Following paternal knock-down of *Hsp83* in the parental (P) generation, a heritable *reduced-eye* phenotype emerged in the F2 generation (Fig. 1a). This phenotype had never been observed before in the Cro1 population.

Initially, parental *Hsp83*-knock-down induced various abnormal phenotypes in F1-larvae, consistent with previous observations of HSP90 inhibition in other species[3] (Extended Data Fig. 1a, c). The eclosed F1-adults frequently exhibited leg malformations (Extended Data Fig. 1b, d). However, none of these traits were heritable, except for a *reduced-eye* phenotype, which was expressed from paternal treatment with an incidence rate of 4.2% (32/757 adult beetles) in F2 offspring produced from pairings of F1-adults with leg malformations (Fig. 1b (i) and 1c). Of 15 such pairings, the phenotype was present in only two families, but at high frequencies of 25.5 and 29.4% (see Extended Data Table 1 for details). The released abnormal phenotypes differed significantly among treatments (Extended Data Fig. 1e (i)).

*Hsp83* knock-down in the injected beetles was confirmed by qRT-PCR (Extended Data Fig. 2a) while *Hsp83* expression was normal in offspring, indicating the depletion of the introduced ds-RNA (Extended Data Fig. 2b). High *Hsp83*-dsRNA disrupted oogenesis, causing female sterility (Extended Data Fig. 2c), so female treatment groups were excluded.

Further, the *reduced-eye* trait continued to be expressed at significant proportions in subsequent generations in the absence of RNAi knock-down of *Hsp83*, consistent with a genetic inheritance of the trait (RNAi-original polymorphic lines, Fig. 1a). Mating of *reduced-eye* beetles in the F5 invariably produced *reduced-eye* offspring, indicating a single recessive locus underlying the phenotype. Using *reduced-eye* individuals from these crossings, we established a monomorphic line fixed for the phenotype (RNAi-monomorphic line, Fig. 1a).

### HSP90 dependent phenotype confirmed by chemical inhibition
To confirm the link between the *reduced-eye* phenotype and HSP90 deficiency, we treated Cro1 larvae with 17-DMAG, a specific inhibitor of HSP90[31] (Extended Data Fig. 3). Quantitative RT-qPCR of the *Hsp68a* gene (a member of the HSP70 protein family), whose expression is expected to increase upon HSP90 inhibition[32,33], confirmed a successful treatment of beetles (Extended Data Fig. 2d). 17-DMAG inhibits HSP90 at the protein level by binding to its ATP-binding pocket, thereby blocking its chaperone activity without affecting *Hsp83* mRNA levels[31,34]. As a result, upregulation of HSP70 family genes, such as *Hsp68a*, serves as a molecular marker of successful HSP90 inhibition[32]. Pairings of beetles treated with 17-DMAG during larval development resulted in a *reduced-eye* phenotype in F1 offspring, with an overall incidence of 0.4% (1/226) at low concentrations (10 μg/mL) and 5.1% (39/764) at high concentrations (100 μg/mL) (Fig. 1b (ii) and 1c; trait incidence rate per family is shown in Extended Data Table 1).

The original *reduced-eye* phenotypes resulting from the RNAi and 17-DMAG experiments were identical, characterized by a strong reduction in eye size associated with a decrease in the number of ommatidia (Fig. 1c, Extended Data Fig. 2e). Mono- and polymorphic 17-DMAG lines were then established for further study (Extended Data Fig. 3a). We further independently repeated the 17-DMAG experiment twice (Extended Data Fig. 3b, c), observing the same *reduced-eye* phenotype in F2 at frequencies of 0.6% (12/1905) and 0.98% (7/712) at 17-DMAG high concentration (Fig. 1b (iii) and (iv); incidence rate per family is shown in Extended Data Table 1). The *reduced-eye* beetles from both the RNAi- and 17-DMAG- monomorphic lines showed no significant difference in *Hsp83* expression compared to naive beetles (Extended Data Fig. 2f), suggesting that the phenotype had become independent of ongoing *Hsp83* expression.

### Development and fitness of the *reduced-eye* phenotype
To further characterize the *reduced-eye* phenotype throughout development, we used the monomorphic lines—after crossing with Cro1 wildtype beetles—to compare eye morphology in the produced *reduced-* and *normal-eye* beetles. Because individuals were tracked throughout development, their adult phenotype could be retrospectively assigned to their larval and pupal stages. The stage-normalized eye surface area as well as the relative eye size were significantly lower in *reduced-eye* individuals compared to those with normal eyes, within each developmental stage (Fig. 1d and Extended Data Fig. 4a; Mann-Whitney U test / T-test: $p < 0.01$). While substantial variance in larvae and pupae led to some overlap of the two phenotypes, eye size was clearly distinct in adults, with reduced eyes reaching only half the size (44%) of normal eyes. Accordingly, the number of ommatidia in *reduced-eye* beetles was significantly reduced by ~75% compared to *normal-eye* beetles (Fig. 1e; T-test: $t = 21.55$, $df = 14$, $p < 0.001$). There was no significant difference in normalized head- and body area between the eye phenotypes, except for a small difference in head area at adults stage (Extended Data Fig. 4b, c; head area

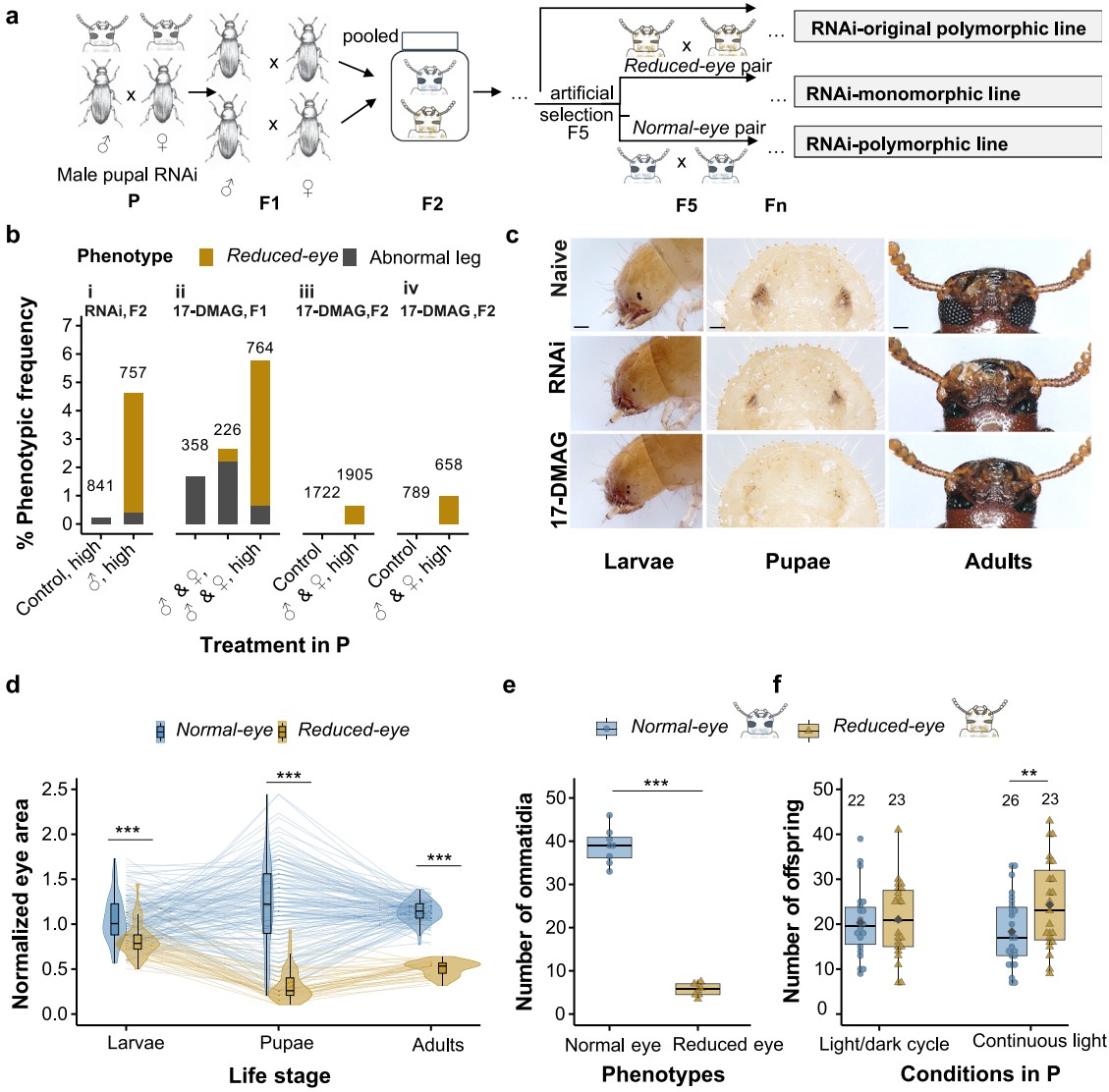

**Fig. 1 | Release of *reduced-eye* phenotype after HSP90 inhibition. a** Eluciding diagram showing *Hsp83* knock-down using RNA interference (RNAi) in a male (♂) pupa of *T. castaneum* releasing a *reduced-eye* phenotype in F2 (shown in brown) followed by establishment of lines, denoted RNAi-lines henceforth. *Normal-eye* phenotype individuals are shown in blue. **b** Percentage of abnormal phenotypes showing the release of the *reduced-eye* trait after HSP90 impairment in the parental generation via RNAi or 17-DMAG (performed three times). RNAi was induced by injecting male pupae with 100 ng/μL dsRNA ("♂, high") followed by crossing with naive beetles; as control we used *AsnA*-dsRNA. 17-DMAG was applied at 10 μg/mL (low) and 100 μg/mL (high), with treated males and females crossed (♂ & ♀); control indicates food without 17-DMAG. Total screened individuals per treatment are shown above each bar. Complete data on all used concentrations and crossing combinations are shown in Extended Data Fig. 1e. **c** *Reduced-eye* phenotype released after RNAi and 17-DMAG treatment in P (middle and bottom rows) compared to naive Cro1 beetles (top row). Three developmental stages are shown: larvae, pupae, and adults. Scale bar: 100 μm. Lateral views for pupae and adults are provided in Extended Data Fig. 1f. **d** Normalized eye area in *normal-* and *reduced-eye* individuals across the three developmental stages, showing significant differences in eye size

within each stage. Eye areas were normalized by the stage mean. Lines connect individual beetles tracked from larval to adult stage. Phenotypes (*normal-* vs. *reduced-eye*) were assigned retrospectively based on adult morphology ($n = 156$ normal-eye and $n = 44$ reduced-eye beetles). Two-sided Mann–Whitney U tests were performed for larval ($p = 5.18 \times 10^{-9}$) and pupal ($p = 4.87 \times 10^{-21}$) stages, and a two-sided t-test for adults (t = 33.4, df = 198, $p = 2.63 \times 10^{-83}$). **e** Average number of ommatidia in *normal-* and *reduced-eye* adult beetles ($n = 8$); each dot represents a single beetle. a two-sided t-test was performed ($t = 21.55$, df = 14, $p = 3.91 \times 10^{-12}$). **f** Number of adult offspring from parents kept under standard light/dark cycle or continuous light conditions, showing fitness consequences of *reduced-eye* phenotype. Numbers above bars indicate total families per treatment; each dot represents one family. Dark gray diamond = mean. Under continuous light, a generalized linear mixed model (negative binomial) with family as a random effect detected a significant difference ($z = 3.35$, SE = 0.08, $p_{adj.} = 0.003$). Box plots for (**d**–**f**) show the median (center line), 25th–75th percentile (box bounds), and whiskers extending to the minimum and maximum values within 1.5× the interquartile range. Significant differences indicated by asterisks, ** $p < 0.01$, *** $p < 0.001$. Source data are provided via Zenodo and linked in the Data Availability section.

in adult stage between the phenotypes: T-test: $t = 2.37$, df = 198, $p = 0.02$).

It has repeatedly been brought forward that most cryptic morphological variants buffered by HSP90 are deleterious[16,22,26]. However, the adaptive value of trait innovations depends on the prevailing environmental conditions. To assess the performance and fitness of

*reduced-eye* beetles, we quantified their reproductive output under continuous light stress, a situation that could be encountered by these beetles in their human-commensal environment. We compared offspring numbers between *reduced-* and *normal-eye* pairs derived from crosses between Cro1 and the 17-DMAG-monomorphic line (Exp. design: Extended Data Fig. 5). We established these pairs such that

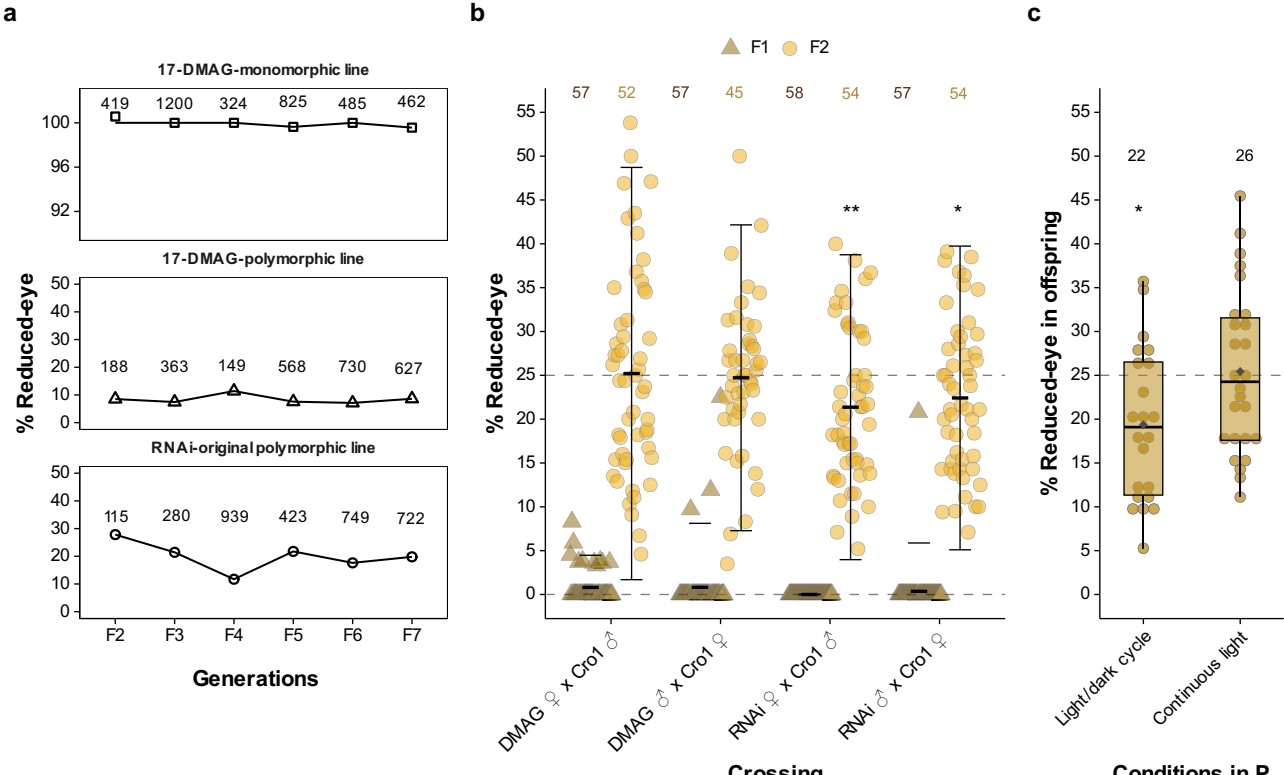

**Fig. 2 | Inheritance of the HSP90-regulated *reduced-eye* phenotype.**
**a** Inheritance of the *reduced-eye* trait across successive generations in established lines derived from both RNAi and 17-DMAG treatments, showing stable transmission of the trait over generations; *numbers* represent the total number of individuals screened per generation. The establishment of the RNAi line is shown in Fig. 1a, and 17-DMAG lines are detailed in Extended Data Fig. 3. **b** Mendelian inheritance of the *reduced-eye* trait in F1- and F2 after outcrossing *reduced-eye* males (♂) and females (♀) from 17-DMAG and RNAi monomorphic lines with wild type Cro1 beetles. Horizontal dashed gray lines indicate 0% and 25% expected Mendelian ratios. Crossbars and error bars represent the mean ± standard deviation for each generation; *numbers represent the* total number of families per treatment; each dot represents one family. RNAi lines showed significant deviations from the expected Mendelian ratio in the F2 generation (Chi-square goodness-of-fit test: RNAi ♀ -

Cro1 ♂, $\chi^2 = 9.35$, df = 1, $p = 0.002$; RNAi ♂ - Cro1 ♀, $\chi^2 = 4.96$, df = 1, $p = 0.026$).
**c** Penetrance of the *reduced-eye* phenotype in offspring from heterozygous normal-eyed parents maintained under standard light/dark or continuous light conditions. Horizontal dashed gray line indicates the expected 25% Mendelian ratio. Each dot represents one family; *numbers represent the* total number of families per treatment. Boxplots show the median and interquartile range (25th–75th percentile), and whiskers extend to the minimum and maximum values within 1.5× the interquartile range. The dark gray diamond represents the mean. In a light/dark cycle, *reduced-eye* offspring were significantly less frequent than expected (19.4% vs. 25; Chi-square goodness of fit: $\chi^2 = 5.76$, df = 1, $p = 0.016\%$), while in continuous light, they matched the Mendelian expectation (25.4%). Significant differences indicated by asterisks, *$p < 0.05$, **$p < 0.01$. Source data are provided via Zenodo and linked in the Data Availability section.

each family produced sibling pairs of both the phenotypes, ensuring same genetic background. Pairs from each family were then distributed across the continuous light and standard light/dark conditions. Under continuous light conditions, *reduced-eye* pairs produced significantly more offspring (mean = 24.3) than *normal-eye* pairs (mean = 18.3) (Fig. 1f; GLMM: $z = 3.35$, SE = 0.08, $p_{adj.} = 0.003$), indicating that *reduced-eye* beetles performed significantly better under light stress than normal-eye ones. Meanwhile, no significant differences were observed in offspring number under light/dark cycle (mean *reduced-eye* pairs: 21.0, *normal-eye* pairs: 20.4; EMMs contrasts after GLMM: $z = -0.27$, SE = 0.09, $p_{adj.} = 0.78$). Together, these results provide evidence for the adaptive advantage of the HSP90-buffered and formerly cryptic *reduced-eye* phenotype under light stress. Developmental speed did not differ significantly between both types of beetles (time to pupation: hazard ratio = 1.19, SE = 0.13, $p = 0.18$; time to eclosion: hazard ratio = 1.20, SE = 0.13, $p = 0.16$; Extended Data Fig. 4d), consistent with the absence of developmental delay caused by the *reduced-eye* variant.

## Inheritance of the *reduced-eye* phenotype
To explore patterns of inheritance, we quantified the *reduced-eye* phenotype in monomorphic and different polymorphic lines,

originally established from RNAi and 17-DMAG treatments (Fig. 1a, Extended Data Fig. 3a) over seven generations (Fig. 2a). Both monomorphic and polymorphic lines represent genetically variable populations, containing the *reduced-eye* allele at some, albeit unknown, frequency. In polymorphic lines, the *reduced-eye* phenotype occurred at rates ranging between 10% to 28% with no tendency towards fixation or extinction, consistent with neither detrimental nor positive fitness effects of the *reduced-eye* phenotype at control conditions (Fig. 2a). Surprisingly, the 17-DMAG-monomorphic line putatively fixed for the *reduced-eye* allele showed incomplete penetrance in generations F5 (99.6%, 822 *reduced-eye* beetles out of 825) and F7 (99.6%, 460 out of 462).

To confirm that the *reduced-eye* phenotype produced by *Hsp83* RNAi and the 17-DMAG treatment shares the same genetic basis, we crossed *reduced-eye* beetles from the 17-DMAG monomorphic line with beetles from the RNAi monomorphic line. In all crosses, 100% of the F1 offspring exhibited the *reduced-eye* phenotype, supporting that the same genetic locus underlies this trait in both independently established lines.

To further check patterns of inheritance, *reduced-eye* beetles from the RNAi- and the 17-DMAG-monomorphic lines were crossed with wildtype beetles (Cro1). Overall, inheritance of the *reduced-eye*

phenotype in these crosses mirrored Mendelian inheritance (Fig. 2b). However, we found phenotype frequencies above 0% in the F1, violating strict Mendelian inheritance of the trait. Frequencies in the F1 remained below 25%, so that they could also not be explained by a rare Cro1 beetle carrying the relevant *reduced-eye* allele. In the F2 generation, most families showed the expected Mendelian ratio of 25%. However, in the crosses RNAi ♀ - Cro1 ♂ and RNAi ♂ - Cro1 ♀, we observed a significantly lower proportion of the *reduced-eye* phenotype than the expected 25% (Chi-square goodness-of-fit test: RNAi ♀ - Cro1 ♂, $\chi^2 = 9.35$, df = 1, $p = 0.002$; RNAi ♂ - Cro1 ♀, $\chi^2 = 4.96$, df = 1, $p = 0.026$).

Based on these deviations from strict Mendelian principles, we hypothesized that the penetrance of the *reduced-eye* phenotype is influenced by environmental conditions. To explore environmental effects on the expression of the trait, we compared the proportion of *reduced-eye* offspring produced by presumed heterozygous *normal-eye* pairs (by crossing 17-DMAG monomorphic line beetles with wild type Cro1 followed by a backcrossing with 17-DMAG-monomorphic line beetles; Extended Data Fig. 5). Pairs maintained under continuous light conditions produced a significantly higher proportion of *reduced-eye* offspring than those maintained under light/dark cycle condition (Fig. 2c; GLMM: $z = 1.98$, SE = 0.16, $p = 0.048$), consistent with environmental cues affecting trait expression. Specifically, in light/dark cycle, the observed proportion of *reduced-eye* offspring (mean: 19.4%) was significantly lower than the expected 25% based on Mendelian inheritance (Chi-square goodness of fit: $\chi^2 = 5.76$, df = 1, $p = 0.016$), while it did not deviate from 25% in continuous light (mean: 25.4; Chi-square goodness of fit: $\chi^2 = 0.085$, df = 1, $p = 0.771$).

The trait's deviation from strict Mendelian inheritance and its sensitivity to environmental stress suggest an epigenetic mechanism interacting with genetic factors to regulate the phenotype. To investigate this, we examined the role of histone acetylation[4], a key epigenetic modification, in modulating the *reduced-eye* phenotype. Feeding *reduced-eye* beetles from monomorphic lines with the HDAC inhibitors Sodium Butyrate (NaB) and Trichostatin A (TSA) increased *normal-eye* offspring (Extended Data Fig. 6), implicating histone acetylation in regulating this HSP90-buffered trait. Treatment with NaB (500 mM) in RNAi line achieved a 33.3% reversion of the *reduced-eye* phenotype, while TSA (50 μM) in 17-DMAG resulted in a significant 23.8% reversion (GLM-binomial, $z = -2.95$, SE = 0.72, $p_{adj.} = 0.009$).

### Genetic basis of the *reduced-eye* phenotype

To identify loci associated with the *reduced-eye* phenotype, we sequenced four pooled samples using bulks[35,36], each containing 50 adult beetles (generation F51) representing the *reduced-* and *normal-eye* phenotypes from the RNAi-original and 17-DMAG polymorphic lines (Fig. 1a, Extended Data Fig. 3a). After 51 generations, sufficient recombination events are expected to have occurred between the *reduced-* and *normal-eye* alleles, allowing accurate identification of the genomic region linked to eye phenotype variation.

We tested for significant changes in genome-wide allele frequencies between *reduced-* and *normal-eye* in each line using Fisher's exact test, followed by corrections for multiple testing. The analysis identified a single peak at ~3.9 Mb on linkage group 10 (LG10), showing highly significant allele frequency differences between *normal-* and *reduced-eye* phenotypes in the RNAi and 17-DMAG lines (Fig. 3a). The peaks overlapped in both lines, consistent with a single locus underlying the *reduced-eye* phenotype. Thus, we extracted only SNPs with an FDR-corrected $p < 10^{-20}$ in both lines, yielding 85 candidate SNPs (Fig. 3b, c and Extended Data Table 2).

Based on the position of these SNPs, we initially defined a candidate region of 35.8 kb overlapping five genes (Fig. 3d; left part of the orange area). Since much of this region (Fig. 3d, right part of orange area) and SNPs (Fig. 3c, ~3.9 Mb) lacked annotated genes, we considered the potential involvement of regulatory elements affecting nearby gene expression. To account for this, we expanded the candidate region by 32.1 kb upstream and downstream, yielding a 100 kb region with potentially impacted genes (Fig. 3d). This region contained twelve genes, which were screened for SNP impact, expression profiles, and known roles in developmental processes (Extended Data Table 3).

Six genes were selected for further functional analyses using RNAi knock-down, including TC011336 (a proneural gene called *atonal*, symboled *ato*), which encodes the Atonal protein, involved in the formation of *Drosophila* chordotonal organs and photoreceptors[37,38]. The RNAi-mediated knock-down of these genes resulted in several developmental abnormalities (Extended Data Fig. 7 and Extended Data Table 4). Interestingly, the knock-down of *ato* resulted in a *reduced-eye* phenotype which was indistinguishable from the phenotype seen after HSP90 impairment (Fig. 3e). This phenotype appeared in 51.6% (16/31) of adults, after larval injection with 1000 ng/μL *ato*-dsRNA. Further, in two independent *ato* knock-down experiments, the same eye phenotype was observed in 73.9% (17/23) and 73.4% (138/188) of adults in the second and third experiments, respectively. When a second, non-overlapping dsRNA construct for *ato* was used, the *reduced-eye* phenotype was observed in 41% (48 out of 117) of the insects after the larval injection. Together, these findings suggest that *ato* plays an integral role in the development of this phenotype. Notably, no *reduced-eye* phenotype was observed in the offspring produced from these *reduced-eye* beetles, as expected for larval RNAi experiments. To further prove the link between *ato* expression and the *reduced-eye* phenotype, we used qRT-PCR to compare *ato* expression in our independently established monomorphic lines and found a significant reduction compared to the ancestral Cro1 line (Fig. 3f).

Our mapping analysis shows that the *ato* gene—which consists of a single exon—overlaps with four synonymous polymorphic SNPs across the four pooled samples (Extended Data Fig. 8). However, none of these SNPs was consistently associated with the phenotype when comparing *reduced-* and *normal-eye* beetles within each polymorphic line. These synonymous mutations are unlikely to result in a non-functional Atonal protein and therefore cannot explain the phenotype. This suggests that the phenotype may instead be caused by an as-yet-undiscovered regulatory sequence regulating *ato* expression. We used two publicly available ATAC-seq[39] and FAIRE-seq[40] datasets to explore potential *cis*-regulatory regions. Particularly an area lacking annotated genes, yet enriched in highly differentiated SNPs, showed peaks of chromatin accessibility in both ATAC-seq and FAIRE-seq datasets (Extended Data Fig. 9).

### *Hsp83* knock-down significantly reduces *ato* expression

To further test for a functional link between HSP90 and *ato* gene expression, we performed an RNAi experiment. We injected 14-day-old larvae with 5 ng/μL of *Hsp83*-specific dsRNA and measured gene expression in the head tissue 4 days post-injection. This treatment reduced *Hsp83* transcript levels by ~50% and resulted in a ~70% decrease in *ato* expression (Fig. 3g), indicating that HSP90 positively regulates *ato* expression, thus suggesting a regulatory connection between HSP90 and the *ato* gene.

## Discussion

More than 80 years after Waddington proposed his concept of canalization[1], and 25 years after Rutherford and Lindquist's demonstration of the release of phenotypic diversity upon HSP90 reduction[3], the evolutionary relevance of capacitance in animals remains controversial due to the absence of clearly identified HSP90-regulated genes producing potentially adaptive phenotypes[27,41–43]. With the present study, using the insect model *T. castaneum*[28,29] as a promising system for these debates, we show that HSP90 impairment through RNAi and chemical inhibition revealed a stably inherited *reduced-eye* phenotype. This trait was caused by the *ato* gene and showed

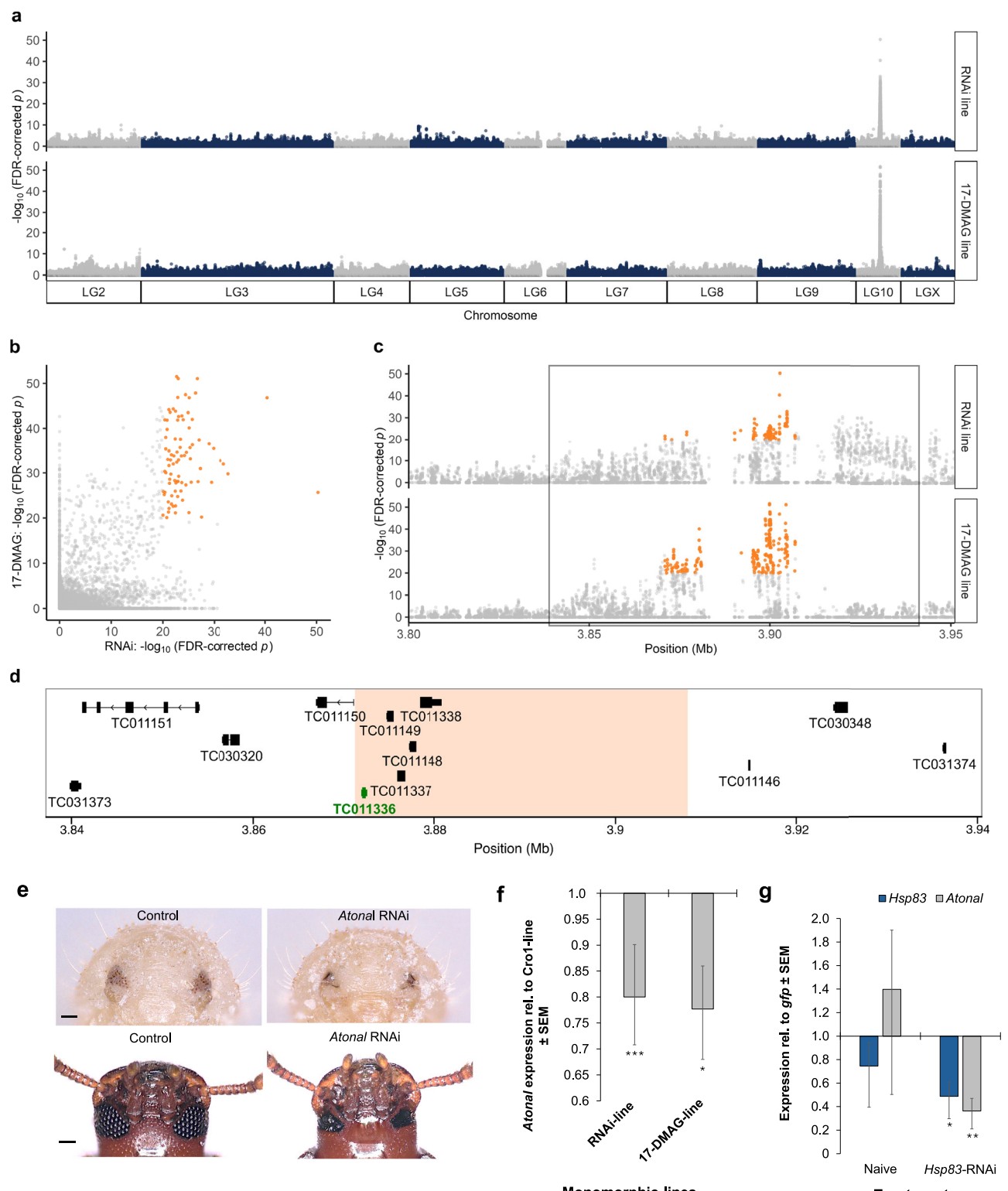

enhanced fitness under continuous light conditions, demonstrating how HSP90 impairment can uncover potentially adaptive genetic variation. It is well established that certain environmental stresses can limit HSP90 availability by increasing the demand for its chaperone function[14], thereby unmasking cryptic genetic variation and producing new phenotypes[3]. Our findings suggest that once such variation is exposed—regardless of the initial stressor—it can become subject to selection in a novel environment, even if that environment does not directly affect HSP90 expression.

The *reduced-eye* phenotype was released in the F2 generation following *Hsp83* RNAi, as well as in the F1 and F2 after 17-DMAG-mediated HSP90 inhibition. These differences likely stem from timing, duration, and mechanisms of action. RNAi transiently disrupts HSP90 in a stage-specific manner, exposing cryptic variation that can trigger heritable epigenetic reprogramming, such as altered DNA methylation or histone modifications, manifesting in the F2 generation[44,45]. In contrast, 17-DMAG feeding for 11 days systemically inhibits HSP90 across all tissues, including germline cells,

**Fig. 3 | Genomic basis underlying eye phenotype variation. a** Manhattan plot showing genome-wide allele frequency differences between *normal-* and *reduced-eye* phenotypes in the RNAi (top) and 17-DMAG (bottom) lines. The *Y* axis represents the $-\log_{10}$ FDR-corrected *p*-values for each SNP based on two-sided Fisher's exact test (FET) implemented in PoPoolation2, while the *X* axis depicts the ten *T. castaneum* linkage groups (LG). In both RNAi and 17-DMAG lines, a single peak at 3.9 Mb on LG 10 is associated with the most significant SNPs. For sequencing, DNA was extracted from four pooled samples ($n = 50$ per pool): one *reduced-eye* and one *normal-eye* pool from each line (RNAi and 17-DMAG). **b** Correlation between $-\log_{10}$ FDR-corrected *p*-values in the RNAi and 17-DMAG lines, with candidate SNPs ($-\log_{10}$ FDR-corrected *p*-value $\geq 20$ in both lines) highlighted in orange. Values are derived from the same FET analysis shown in panel a. **c** Zoomed-in view of the candidate region showing candidate SNPs (highlighted in orange) and the extended 100 kb candidate region (black rectangle). **d** Gene annotations for the twelve genes overlapping the 100 kb candidate region, with *ato* (TC011336) shown in green.

The orange rectangle highlights the initial 35.8 kb candidate region overlapping five genes including the *ato* gene. **e** *Reduced-eye* phenotype in pupae (top) and adults (bottom) after larval *ato* knock-down using RNAi with an injection of 1000 ng/μL *ato*-dsRNA. Scale bar: 100 μm. **f** *Ato* relative expression in larval head tissue from individuals randomly selected from RNAi- or 17-DMAG monomorphic lines. Three biological replicates (25 larval heads each) were used. REST: RNAi-line, $p = 0$; 17-DMAG-line, $p = 0.033$. **g** Relative expression of *Hsp83* and *ato* after *Hsp83* knock-down showing a significant reduction in *ato*, supporting a regulatory link. Four biological replicates (25 larval heads each) were analyzed 4 days after injection of 5 ng/μL of *Hsp83*-specific dsRNA; *gfp* was used as a control. For *Hsp83* RNAi treatment: REST $p = 0.021$ (*Hsp83*); $p = 0.006$ (*Ato*). For **f, g** bars show mean relative expression calculated from REST output processed in Excel, +/- standard error of the mean (SEM). Significant differences indicated by asterisks, *$p < 0.05$, **$p < 0.01$, ***$p < 0.001$. Source data are provided via Zenodo and linked in the Data Availability section.

destabilizing a broader range of client proteins and regulatory networks. This stronger, global disruption likely explains the immediate F1 phenotypic expression[46,47]. A recent study showed that 17-DMAG inhibition of HSP90 had a stronger effect than siRNA knock-down on MET, an HSP90 client protein[48]. The sustained inhibition by 17-DMAG more effectively disrupts developmental buffering, amplifying epigenetic and genetic destabilization[49].

Rutherford and Lindquist's original idea was that HSP90 serves as an evolutionary capacitor due to its function as a protein chaperone. Alternatively, HSP90 inhibition might also produce novel genetic variants through the activation of TEs[19,21]. The important difference between these two concepts lies in the fact that according to Rutherford and Lindquist, genetic variants that have been proven successful under certain conditions can remain in a population's gene pool, whereas the vast majority of new genetic variants produced by an HSP90 mutator effect through TEs activity are likely to be harmful. Additionally, HSP90 may regulate epigenetic variance that can be assimilated into the population by selection over multiple generations[4].

Our data supports this last option, for the following reasons. Crosses between *reduced-eye* beetles from RNAi- and 17-DMAG-monomorphic lines indicated that the same recessive locus underlies this phenotype. The likelihood of TEs inserting into the same gene or regulatory region in repeated experiments is low, whereas a cryptic *reduced-eye* allele in the starting population could explain the recurrent phenotype. Accordingly, our genetic analyses identified the same region on linkage group 10, including *ato*, a proneural gene crucial for photoreceptor development[37,38]. Since eye differentiation depends on precise *ato* expression, regulatory variation in this region may affect its spatial or temporal activity[50]. RNAi knock-down of *ato* consistently induced a phenotype identical to that observed following HSP90 impairment and thus provides strong evidence for its involvement. Moreover, *ato* expression was reduced in our independently established, monomorphic *reduced-eye* lines. The *ato* gene lacked nonsynonymous (*i.e.*, protein sequence–changing) polymorphisms, suggesting an involvement of nearby regulatory sequences controlling its expression. RNAi-mediated knock-down of the *Hsp83* gene led to reduced expression of *ato*, supporting a role of HSP90 in regulating *ato* expression. Consistent with this hypothesis, recent research in *Sacharomyces cerevisiae* demonstrated that HSP90 predominantly mediates the phenotypic effects of cryptic variation in *cis*-regulatory sequences rather than in protein-coding sequences[20].

HSP90 influences epigenetic regulation by promoting RNA polymerase II pausing near gene promoters and activating paused genes in response to stimuli[51]. It maintains active chromatin at gene expression sites through interaction with Trithorax, stabilizing key developmental genes. This interaction may drive epigenetic inheritance via differential histone mark segregation[18,45,52,53]. Similar phenotypes from Trithorax mutations and HSP90 inhibition in *D. melanogaster* suggest

shared mechanisms in revealing morphological variation[4]. HSP90 also affects DNA binding of bHLH transcription factors like *ato*[54,55]. Our findings suggest a recessive, HSP90-buffered *cis*-variant in *ato* was unmasked upon HSP90 inhibition. Five observations in our study strongly suggest an involvement of epigenetic modulators of the *reduced-eye* phenotype. First, the *reduced-eye* phenotype was originally released from a treated grandfather (Fig. 1b (i)), raising the possibility of transgenerational epigenetic inheritance via the male germline[44,45]. Second, the inheritance of the phenotype deviates slightly from expected Mendelian ratios, suggesting it is not strictly determined by classical genetic inheritance (Fig. 2b). Third, fluctuating penetrance of the trait in the 17-DMAG-monomorphic line – despite expectations of 100% expression if the trait would be purely genetic (e.g., homozygous recessive)–points to non-genetic influences (Fig. 2a). Fourth, the enhanced fitness and expression of the trait under continuous light condition underscores the role of environmental factors in modulating its penetrance (Figs. 1f and 2c). Fifth, the reversion of the *reduced-eye* phenotype to the normal state after using the HDAC inhibitors NaB and TSA in both monomorphic lines indicates that HSP90 epigenetically buffers the *reduced-eye* phenotype through stabilizing histone acetylation level (Extended Data Fig. 6).

Together, these findings implicate epigenetic processes potentially mediated by HSP90, as a significant factor in regulating the *reduced-eye* phenotype. This is consistent with similar mechanisms reported in other organisms, where epigenetic regulation of HSP90-buffered traits occurs through chromatin modifications or DNA methylation[4,54,56,57]. In *Drosophila*, traits expressed following HSP90 reduction were reversed to the wild-type state with HDAC inhibitors such as NaB and TSA[4]. HSP90 plays a crucial role in HDAC3-dependent gene transcription[58]. HDAC inhibitors, including NaB and TSA, exert their effects by binding to the zinc-binding sites of class I and II HDACs, leading to the accumulation of histone acetylation[59].

Contrary to concerns that evolutionary capacitance might not work because phenotypes emerging after HSP90 inhibition are typically deleterious[22,23,26], our study provides evidence that the *reduced-eye* phenotype is not detrimental. It remained at stable frequency in unselected populations under normal lab conditions (Fig. 2a) and even confers fitness benefits under continuous light (Figs.1f and 2c). This aligns with recent studies in yeast and other fungi, where HSP90 has been suggested to potentiate adaptation[20,24,25,60–62], as well as studies in cavefish, where HSP90-mediated variation in eye-size might have facilitated eye loss as an adaptation to cave conditions. Insects display remarkable diversity in the size and shape of their compound eyes, enabling them to adapt to a wide range of environments and lifestyles[63–65]. As a stored-product pest, *T. castaneum* beetles are photonegative, *i.e.*, attracted to dark environments[66–68]. Developing fully functional, large eyes is energetically expensive[69–71]. Smaller eyes might also avoid light-induced stress[72]. Accordingly, reduced eyes are common in cave-dwelling species[73]. Interestingly, eye phenotypes were

affected by HSP90 inhibition in several species[3,12,74], raising the possibility that the buffering by HSP90 is directed towards certain morphological traits. The role of *Hsp83* in *T. castaneum* eye development has been examined in a single study, which focused on effects within a single generation[75].

In a previous study in *T. castaneum*, HSP90 downregulation was observed in a risky environment, *i.e.*, in cohabitation with wounded conspecifics[17]. Although this regulatory response was not directly addressed in the present study, the two studies together suggest a scenario that links evolutionary capacitance–and thus evolvability of a population–to environmental conditions. Further studies are needed to determine the relevance of this process, particularly under ecological conditions where rapid adaptation based on standing genetic variation is required, such as coevolution, insecticide resistance, or climate change.

## Methods

### Model organism
Stocks of the red flour beetle, *Tribolium castaneum*, (Coleoptera: Tenebrionidae) (Cro1 strain) were maintained as non-overlapping generations under standard rearing conditions (30 °C and 70% relative humidity with a 12 h light/dark cycle) in wheat flour with 5% brewer's yeast[30]. This beetle strain was collected in Croatia in 2010 and allowed to adapt to laboratory conditions in large outbreeding populations for at least 20 generations before starting any experiments[30].

### RNAi-mediated *Hsp83* knock-down
In our study, we focused on *Hsp83*, a highly expressed gene encoding a cytosolic protein of the HSP90 family and homolog of *Hsp83* in *Drosophila*. It differs from its paralogue, *Hsp90*, which codes for an endoplasmic reticulum-based HSP90 protein. knock-down of *Hsp83* was achieved by synthesizing *Hsp83* double-stranded RNA (dsRNA). Initially, *Hsp83*-specific primers were designed (https://www.metabion.com/) targeting the coding sequence (CDS) region spanning nucleotides 508–964 (457 bp) of the Tc-*Hsp83* gene. The amplified *Hsp83* fragment was cloned into the pZERrO™-2 vector (Invitrogen, Catalog number 45-0448) to add T7 and T7-SP6 promoter sequences, and dsRNA was subsequently synthesized using a T7 MEGAscript Kit (Ambion, Catalog number AMB1334-5) (sequences shown in Extended Data Fig. 10).

RNAi was performed following established protocols[76] by injecting 3-day-old male and female pupae, which were randomly selected from naive Cro1 stock[76]. Two concentrations of *Hsp83*-dsRNA were used: 20 ng/μL (low) and 100 ng/μL (high). Before injection, pupae were individualized during the larval stage to prevent cannibalism, ensuring the absence of morphological variation. As RNAi control, dsRNA specific for *asparagine synthetase A* (*AsnA*)[77], a gene specific to *Escherichia coli*, was used. Injected pupae were maintained individually in 5% yeasted flour under standard rearing conditions until eclosion.

### Morphological variation in subsequent generations following parental *Hsp83* knock-down
Parental *Hsp83*-RNAi was performed by crossing the eclosed beetles (5 day old) as single pairs (n = 20-30) in three mating regimes: (1) Male (symbolized in figures as ♂): only males were treated with one of the dsRNA concentrations (low or high) and crossed with naive females; (2) Female (♀): only females were treated and crossed with naive males; and (3) Male and Female (♂ & ♀): both partners were treated. After 3 days of egg-laying, untreated, F1-larvae (10 days old) were individualized to prevent cannibalism and inspected morphologically under a binocular microscope 5 days later (Extended Data Fig. 1a, c). The larvae were then re-individualized and monitored until eclosion, with the sample size reduced due to some larvae not reaching adulthood. Larvae exhibiting abnormal phenotypes were kept separately to follow their development into adults. Three weeks later, F1 adults were examined for phenotypic variation, focusing on variations in body size,

coloration, and structural traits of the head, thorax, abdomen, antennae, eyes, mouthparts, legs, and urgomophi (Extended Data Fig. 1b–d). Images of notable traits were captured using a Canon digital camera attached to a Zeiss Axioskop compound microscope. Light anesthesia with $CO_2$ was used to immobilize the beetles during imaging.

### Production of lines with specific traits following parental *Hsp83* knock-down
The produced F1-adult beetles that exhibited morphological abnormalities were selected and mated as single pairs (n = 11–19 pairs, with most showing abnormal leg phenotypes) for each treatment group to generate F2 generations. F2 adults were morphologically examined, with a focus on the trait of interest (i.e., which were seen in F1) and any newly observed traits. Notably, the *reduced-eye* phenotype was not genetically linked to leg malformations, as it also appeared in F2 offspring from normal-leg F1 siblings of the same RNAi-treated family (unpublished data). F2 individuals with specific traits were then mated in groups rather than single pairs to increase the population size and generate sufficient offspring for studying the inheritance of certain morphological variations across generations. To establish an 'RNAi-original polymorphic line', beetles that showed interesting traits in the F2 generation were pooled with their normal siblings, without further artificial selection, to produce the F3 generation. The trait was then monitored until F7 (Figs. 1a and 2a). In the F5, a *reduced-eye* pair was selected to establish an 'RNAi-monomorphic line', while a *normal-eye* pair was chosen to create an 'RNAi-polymorphic line' (Fig. 1a).

### Pharmacological inhibition of HSP90
The water-soluble geldanamycin analogue 17-Dimethylaminoethylamino-17-demethoxygeldanamycin, (17-DMAG; InvivoGEN, product number S1142-5) was used at concentrations of 10 μg/mL (low) and 100 μg/mL (high) to inhibit HSP90 function. Flour disks (thin layers of flour) containing 17-DMAG were prepared following a modified protocol[30]. A solution of 17-DMAG at the desired concentration was mixed with flour (0.25 g/mL), pipetted into 96-well plates (40 μl/well), and dried for 5 h at 30 °C in an air-circulating oven. Larvae (12-16 days old) were randomly selected from naive Cro1-rearing stocks, fed on the prepared flour disks, and kept in the darkness to prevent light degradation of 17-DMAG. Freshly prepared disks were provided every other day for up to six feedings or until pupation. Flour disks without 17-DMAG were used as negative control. Eclosed virgin adults were transferred to standard rearing conditions and fed loose flour for 7 days before single pair crosses were established (n = 6 – 13). Pairs were arranged such that both partners were treated or only one (male or female) was treated, crossed with an untreated Cro1 partner. Eggs were collected after 7 days, and untreated F1 adults were morphologically inspected, as described above. Some pairs produced F1-adult beetles with morphological abnormalities, including a *reduced-eye* phenotype. To maintain a consistent genetic background, we established two distinct 17-DMAG lines from a single pair (Extended Data Fig. 3a). A '17-DMAG-monomorphic line' was created by selecting for the *reduced-eye* phenotype, while a '17-DMAG-polymorphic line' was established by selecting against the *reduced-eye* phenotype (i.e., using *normal-eye* siblings). Both lines were bred under standard conditions without further artificial selection, and the trait was monitored every generation up to F7 (Fig. 2a and Extended Data Fig. 3a). The entire experiment was repeated twice to confirm the findings (Extended Data Fig. 3b, c).

### Real time-quantitative PCR confirming HSP90 impairment
To confirm the potential effect of RNAi and 17-DMAG, we used quantitative real-time polymerase chain reaction (RT-qPCR) analysis following established protocols[17]. RT-qPCR primers for *Hsp83* are shown in Extended Data Fig. 10. For RNAi, *Hsp83* expression was measured in both eclosed adults and untreated larval offspring. Four biological

replicates were analyzed, each consisting of pools of five adult beetles (7 days post-pupal injection) or ten F1 larvae (Extended Data Fig. 2a and b). For 17-DMAG, the expression of *Hsp68a* was used as a molecular marker for HSP90 inhibition after 17-DMAG treatment[32,33].

*Hsp68a* expression was measured in individuals collected at different time points of feeding on flour disks containing 17-DMAG at 100 μg/mL (on days 2, 4, 7, 9, and 11, and on day 18, one week after the last feeding time). Four biological replicates of pools of 8 individuals were used for each time point (Extended Data Fig. 2d).

Furthermore, *Hsp83* and *ato* relative expression were checked in *reduced-eye* beetles, randomly chosen from RNAi- or 17-DMAG established monomorphic lines. *Hsp83* expression was measured in three biological replicates each consisting of whole-body samples from five 10-day-old adult beetles (Extended Data Fig. 2f). *Ato* expression was quantified in three biological replicates, each comprising 25 pooled head samples from 20-day-old larvae (Fig. 3f). In all experiments, gene expression was computed relative to the expression of the housekeeping genes *ribosomal protein L13a* (*RpL13a*) and *ribosomal protein 49* (*rp49*). The qRT-PCR data were analysed using Relative Expression Software Tool (REST) software[78]. Mean Cp values of two technical replicates were used for each sample. When the standard deviation between replicates exceeded 0.5, the replicate closest to the mean Cp of the other respective biological replicates within the treatment group was retained.

## Quantitative analysis of developmental differences in the *reduced-eye* phenotype

To evaluate morphological differences between *reduced-* and *normal-eye* beetles, body area, head area, eye area, eye size, and ommatidia characteristics were measured at each developmental stage (larva, pupa, and adult) (Fig.1d, e and Extended Data Fig. 4a–c). Comprehensive measurements were conducted in F2-beetles produced from crosses between *reduced-* and *normal-eye* beetles (RNAi- or 17-DMAG-monomorphic × Cro1 and Cro1 × Cro1). Both males and females were reciprocally used in these pairings. Measurements were taken from 200 individuals representing both phenotypes (n = 156 *normal-eye* and n = 44 *reduced-eye* beetles) at three developmental stages: larval (17 days), pupal (25 days), and adult (40 days).

Larvae were placed individually in 96-well plates and maintained there until pupation, during which the sex of pupae was determined at 23–24 days post-egg-lay. All beetles were photographed using a Keyence digital microscope (VHX-900F) at 200x for eye imaging and 50x for other body parts. Beetle eyes were photographed with all specimens placed in a fixed position to ensure consistency, using constant measuring parameters. For pupae and adults, images were captured from the ventral side, while larvae were imaged laterally. Measurements –including eye area, head area, and body length– were conducted using the ImageJ program[68]. Body length was calculated as the average of dorsal and ventral measurements from the anterior end of the head to the posterior end of the abdomen. Head area was determined using head width and length, while eye area was measured by isolating the black eye region with ImageJ's color threshold option. In *Tribolium*, the eye has a horseshoe shape, so only the largest visible part is measurable when the beetle is placed ventrally. Brightness and saturation were adjusted for thresholding and area measured in mm². Relative eye size was calculated as the ratio of eye area to head area for each beetle.

The number of ommatidia in adult beetles (n = 8) was calculated as the average of counts from both compound eyes. This measurement was limited to beetles with clearly distinguishable ommatidial structures, as distortions in reduced eyes made accurate counting unreliable.

## Fitness consequences of *reduced-eye* phenotype

*Reproductive success*. We evaluated the fitness effects of the *reduced-eye* phenotype by comparing fecundity between *reduced-* and *normal-eye* siblings (Fig. 1f). To control for genetic background, sibling pairs of both phenotypes were generated from the same families. This was achieved by initially crossing Cro1 beetles (*normal-eye*) with homozygous *reduced-eye* beetles from the 17-DMAG-monomorphic line to produce presumed heterozygous *normal-eye* beetles. These heterozygous individuals were then backcrossed with homozygous *reduced-eye* beetles from the monomorphic line, using single-pair matings (n = 22-26).

Offspring were sexed and individualized at the pupal stage, and their phenotypes were determined after eclosion. Ten-day-old *reduced-* and *normal-eye* beetles from each family were then exposed to either continuous light or a standard 12-h light/ 12-h dark cycle for 24 h prior to and during three days of mating, with a daily transfer to fresh flour (Extended Data Fig. 5). Minimal flour was provided to prevent burrowing and escape from light-induced stress. The total number of offspring reaching the adult stage was recorded as a measure of fecundity.

*Developmental rate of the reduced-eye beetles*. To determine whether the developmental rate differs between *reduced-* and *normal-eye* beetles, we compared the pupation and eclosion (adult emergence) times of individuals with these phenotypes (Extended Data Fig. 4d). A total of 800 larvae, 13 days old, were randomly selected from the 17-DMAG polymorphic line and placed into petri-dishes (50 larvae per dish) containing 5% yeasted flour. The dishes were maintained under a standard light/dark cycle. Larvae were monitored every other day to record pupation. Once pupae emerged, they were sexed and transferred individually into 96-well plates for daily observation of eclosion. The eye phenotype of eclosed adults was determined using a light microscope. This process identified 67 *reduced-eye* and 653 *normal-eye* beetles. The *reduced-eye* phenotype exhibits ~10% penetrance in this line, which accounts for the significant disparity in sample sizes between the two phenotypes. Monitoring of pupation was concluded on the 32[nd] day after oviposition, while eclosion observations continued until the 35[th] day.

## Inheritance of the *reduced-eye* phenotype

*Trait inheritance in monomorphic and polymorphic lines*. As detailed in the earlier sections on 'Hsp83 knock-down' and 'Chemical Inhibition of HSP90', we established beetle lines through RNAi and 17-DMAG treatments, screening all produced individuals across seven generations (Fig. 1a, Extended Data Fig. 3a). To explore segregation patterns in detail, we quantified the inheritance of the trait in these monomorphic and polymorphic lines (Fig. 2a).

*Testing the genetic basis through crosses*. To assess whether the *reduced-eye* phenotype induced by *Hsp83* RNAi and 17-DMAG treatment shares the same genetic determinant or arises from different genetic loci, we conducted crosses between *reduced-eye* beetles from the 17-DMAG- and RNAi-monomorphic lines. Males from one line were paired with females from the other line, and vice versa. Pupae from both lines were randomly selected and paired as adults (10 days old) in single mating setups (n = 6–10). The adult offspring were then examined for the *reduced-eye* phenotype 30 days after egg laying, offering insights into whether this trait is governed by the same genetic locus in both independently derived lines.

*Mendelian inheritance*. To determine whether the heritable *reduced-eye* phenotype follows Mendelian inheritance, outcrosses with naïve Cro1 beetles were performed (Fig. 2b). Using monomorphic lines, four cross combinations were established: (1) RNAi-monomorphic ♂ crossed with Cro1 ♀, (2) RNAi-monomorphic ♀ crossed with Cro1 ♂, (3) 17-DMAG-monomorphic ♂ crossed with Cro1 ♀, and (4) 17-DMAG-monomorphic ♀ crossed with Cro1 ♂. Within each group, 10-day-old adult beetles were paired in single crosses and allowed to lay eggs for 3 days in mating vials under standard rearing condition. The emerged pupae were sexed and individualized, and once they eclosed into 10-day-old adults, their eye phenotypes were

assessed. Single pairs ($n = 45$–58) were then mated to produce the next generation. The adult beetles from subsequent two generations were examined for the *reduced-eye* phenotype.

*Environmental influence on the expression of reduced-eye phenotype*. The penetrance of the *reduced-eye* phenotype under two environmental conditions: a standard 12-h light/ 12-h dark cycle and continuous light was investigated. We checked the *reduced-eye* phenotype in the offspring produced from the *normal-eye* pairs ($n = 22$-26), as described in the previous section on fitness consequences under the two environmental conditions (Extended Data Fig. 5, Fig. 2c).

*Effect of the HDAC inhibitors NaB and TSA on the expression of reduced-eye phenotype*. *Reduced-eye* beetles were randomly selected as pupae from RNAi- and 17-DMAG-monomorphic lines. They were sexed, individualized into 96-well plates, and kept under standard conditions until eclosion. Five-day-old adult beetles were fed on flour disks containing Sodium butyrate (NaB, Sigma-Aldrich, product number B5887) at concentrations of 10 and 100 mM for a period of nine days, with the NaB replaced every three days. Two days after the final feeding, the beetles were paired for single-mating crosses ($n = 10$) and allowed to lay eggs for three days. One month later, offspring was examined for the *reduced-eye* phenotype in the adult stage (Extended Data Fig. 6a). The same experiment was repeated in the same manner with a higher sample size ($n = 15$) and by using additional concentrations of NaB, 100 and 500 mM, and the HDAC inhibitor Trichostatin A (TSA, Sigma-Aldrich, product number T8552) at $50 \, \mu$M (Extended Data Fig. 6b).

## Genetic basis of the *reduced-eye* phenotype
*Establishment of bulks*. We used the 17-DMAG- polymorphic and RNAi original-polymorphic lines for Bulk Segregation Analysis (BSA)[35,36]. First, we assessed the usability of these lines to confirm they retained a stable inheritance rate for the *reduced-eye* phenotype, consistent with the pattern shown in Fig. 2a. Phenotypic examination revealed consistent inheritance rates, with 11.04% *reduced-eye* individuals in the 17-DMAG line and 24.02% in the RNAi line. Egg laying was carried out using both F50 lines to produce offspring.

*DNA extraction and Sequencing*. High-molecular-weight DNA was extracted from four pooled samples of $F_{51}$ adult *T. castaneum* beetles ($n = 50$ individuals per pool) using the Kopenhagen protocol. These samples consisted of one pool of *reduced-* and one pool *normal-eye* from the 17-DMAG polymorphic line and one *reduced-* and one *normal-eye* pool from the RNAi polymorphic line. The Kopenhagen protocol involved freezing individual beetles in liquid nitrogen, grinding them with a plastic pestle, and lysing the tissue in salting out buffer (660 μL) supplemented with 10% SDS and Proteinase K. Samples were incubated overnight at 56 °C to ensure complete tissue digestion. DNA purification steps included phenol-chloroform extraction, RNase A treatment, and ethanol precipitation, yielding high-quality DNA suitable for downstream applications.

Paired-end 150-bp sequencing was performed on a NextSeq 2000 system at the Core Facility Genomics, University of Münster. Sequencing was conducted to an average coverage of 300X across all samples (Extended Data Table 5), ensuring sufficient depth for BSA.

*Read Mapping*. After assessing the raw read quality using *FastQC* (version 0.11.7) (https://www.bioinformatics.babraham.ac.uk/projects/fastqc/), we filtered out short and low-quality reads as well as adapter contamination using *Trimmomatic* (version 0.38)[79] with the options: ILLUMINACLIP:TruSeq3-PE.fa:2:30:10:2:keepBothReads SLIDINGWINDOW:4:20 MINLEN:40. The filtered reads were aligned to the *T. castaneum* reference genome (Tcas5.2; GenBank accession: GCA_000002335.3) using *BWA-MEM* (version 0.7.17)[80] with default settings. The alignment mapping quality was evaluated using *Quali-Map* (version 2.2.1)[81] (Extended Data Table 5). Finally, alignment files were coordinate-sorted with *SAMtools' sort* function (version 1.13)[82],

and duplicate reads were marked using *Picard's MarkDuplicates* (version 2.20.0) (https://broadinstitute.github.io/picard/).

*Fisher's Exact Test Estimation*. To calculate Fisher's exact test (FET) on allele frequency differences for each SNP position between *reduced-* and *normal-eye* samples in each polymorphic line, we first combined the four alignment files into a single mpileup file using *SAMtools' mpileup* function. This file was then converted into a synchronized format using PoPoolation2's *mpileup2sync.jar*[83]. FET was calculated using the *fisher-test.pl* script available under PoPoolation2, with the following options: --min-count 4 --min-coverage 20 --max-coverage 2% (equivalent to 384 for RNAi_*normal-eye* sample, 332 for RNAi *reduced_eye*, 387 for 17-DMAG *normal_eye*, and 310 for 17-DMAG *reduced_eye*) --suppress-noninformative. The genome-wide distribution of the $-\log_{10}$ FDR-corrected *p*-values associated with allele frequency differences between *reduced-* and *normal-eye* samples in each line was visualized in R (version 4.3.2)[84].

*SNP calling and annotation*. For single nucleotide polymorphism (SNP) calling and annotation, joint-variant calling was performed using GATK's *haplotypecaller* (version 4.1.2.0) on the duplicates-free alignment files with default parameters across the four samples[85,86]. Given that we sequenced pooled samples, we called the two most likely alleles per site. We excluded indels from the initial raw call set (6,484,038 variants) and the resulting 4,842,648 SNPs were filtered using VCFtools (version 0.1.16)[87] to include only those genotyped across all four samples, with a mean coverage between 47.8X and 231X and a minimum five reads per genotype, resulting in 4,190,054 SNPs. These thresholds were chosen based on visual inspection of the distributions of these parameters in the raw SNP calls. The impacts of the filtered SNPs on protein-coding genes were annotated using SnpEff (version 5.2e)[88] with the genome and gene annotations (Tcas5.2; GenBank accession: GCA_000002335.3).

*Manual screening of candidate genes*. To select genes for functional analysis with RNAi knock-downs, we first inspected whether the twelve genes were associated with high impact SNPs on IGV (Extended Data Table 3). Additionally, the twelve candidate genes were screened for their expression patterns using NCBI's Genome Data Viewer (https://www.ncbi.nlm.nih.gov/gdv/) and RNA-seq datasets across various developmental stages and tissues. We also explored putative known functions and reported RNAi phenotypes for each gene on the iBeetle-Base database (https://ibeetle-base.uni-goettingen.de).

To identify potential regulatory motifs within the candidate region, we used the MEME Suite[21]tps://meme-suite.org) to perform de novo motif discovery. We used the full 100 kb candidate region as input and ran the analysis with default parameters.

## Knock-down of candidate genes, including *ato*, using RNAi
For simplicity, throughout this article, '*ato*' will refer to the *atonal* gene and "Atonal" to the protein, regardless of the organism being discussed. To investigate whether candidate genes are associated with the *reduced-eye* phenotype, RNAi-mediated gene knock-down was performed in both larvae and female pupae in two experimental blocks (Extended Data Table 4). Larvae (18–19 days old, $n = 22$–55) and female pupae (23–25 days old, $n = 40$–57) were injected with 1000 ng/μL of dsRNA targeting each candidate gene. The dsRNA was ordered from Eupheria Biotech (https://www.eupheria.com/, *ato* catalog number iB_09565) based on accession numbers from iBeetleBase (*ato* Gene identifier TC011336). Naive beetles and those injected with green fluorescent protein (*gfp*) dsRNA were used as controls. Following injection, the insects were maintained in petri dishes containing 5% yeasted flour. RT-qPCR subsamples ($n = 16$ individuals) were collected, 2–3 days post-larval injection and 10 days post-pupal injection, and frozen. To confirm the phenotypic effects of *ato* knock-down, a second dsRNA construct was used. However, we were unable to assess knock-down efficiency with this construct because the qPCR primers overlapped its sequence. All sequences for the used constructs and

RT-qPCR primers are provided in Extended Data Fig. 10. Eclosed adults (35 days old) were assessed for survival and eye phenotype. Adults that emerged from the larval treatments were paired for single matings to examine their offspring, while adult females emerging from pupal injections were crossed with Cro1 males.

### Effect of *Hsp83* RNAi knock-down on *ato* expression
To test whether *ato* expression is regulated by *Hsp83*, RNAi was performed by injecting 14-day-old larvae with *Hsp83*-specific dsRNA. The dsRNA was synthesized by Eupheria Biotech (https://www.eupheria.com/, catalog number iB_02310) using sequence information obtained from iBeetleBase (*Hsp83* Gene identifier TC014606), and was injected at a concentration of 5 ng/µL. A total of four biological replicates were generated, each consisting of pooled head tissue from 25 larvae collected 4 days post-injection. Naive (non-injected) larvae and those injected with dsRNA targeting *gfp* were included as negative controls. Following injection, larvae were maintained individually in 96-well plates containing 5% yeasted flour under standard laboratory conditions. Total RNA was extracted for quantitative expression analysis via RT-qPCR to assess *Hsp83* knock-down efficiency and *ato* transcript levels.

### Statistical analyses
The qRT-PCR data were analysed using REST software[78]. All other statistical analyses were performed in R (version 2024.04.2 + 764) using RStudio. All models used in this study are summarized in Extended Data Table 6. As a base, linear model was fitted to the data using the lm function from the stats package[84]. Residuals from the model were extracted and assessed for normality using both visual (histogram and Q-Q plot) and statistical (Shapiro-Wilk test) methods.

*Abnormal phenotypes following HSP90 impairment.* For RNAi and 17-DMAG experiments (Extended Data Fig. 1), the effect of parental treatment (independent variable) on the proportion of abnormal phenotypes in subsequent generations (response variable, percentage) was evaluated. Since residuals significantly deviated from normality, non-parametric tests were applied. The Kruskal-Wallis test was used to assess differences across treatment groups. When significant ($p < 0.05$), pairwise comparisons were performed using Wilcoxon rank-sum tests with Holm correction to adjust for multiple comparisons. Data preprocessing and statistical analyses were conducted using the dplyr package[89].

Families producing fewer than 10 offspring were excluded to ensure robust statistical power. The following adjustments were made: RNAi-F1 larvae: 9 families excluded; average family size after exclusion: 40.9; range: 18–30. RNAi-F1 adults: 19 families excluded; average family size: 38.46; range: 11–24. RNAi-F2: 1 family excluded; average family size: 45.2; range: 11–19. 1st 17-DMAG: 1 family excluded; average family size: 49.4; range: 6–13. 2nd 17-DMAG: No families excluded; average family size: 114.1; range: 14–17. 3rd 17-DMAG: 5 families excluded; average family size: 47.5; range: 13–15.

*Inheritance of the reduced-eye trait across successive generations.* Percentage of the *reduced-eye* for the established lines derived from both RNAi and 17-DMAG was plotted using ggbreak package[90].

*Eye size across developmental stages.* The eye area (Fig. 1d), head area, body area, and relative eye size (calculated as the ratio of eye area to head area, Extended Data Fig. 4) were measured. Values were normalized by dividing each measurement by the mean of the respective value within the corresponding developmental stage. Since no significant differences in normalized eye area were found between sexes or genetic crosses across most stages, data were pooled for analysis. Measurement values represented the response variables, and eye phenotype (*normal* vs. *reduced*) was the predictor variable. Based on data normality, Mann-Whitney U tests or t-tests were performed at each developmental stage.

*Ommatidia number.* A two-sample t-test from the stats package was used to compare Average Ommatidia Numbers between the *reduced-* and *normal-eye* groups (Fig. 1e).

*Effect of reduced-eye phenotype on reproductive success.* Using a Generalized Linear Mixed Model (GLMM), we analysed the number of offspring produced by *reduced-* and *normal-eye* parents under the standard light/dark cycle and continuous light stress (Fig. 1f). We employed a negative binomial regression model to examine the relationship between the number of adult offspring and treatment defined as the light condition-phenotype combination (fixed effect with four levels). To account for individual variability in the response variable across different families, we included family number (family id) as a random intercept. This model was implemented using the glmer.nb function from the lme4 package[91]. Since the specific comparison between phenotypes in the light/dark cycle was not directly tested in the model, we conducted post-hoc pairwise comparisons using Estimated Marginal Means (EMMs) with the emmeans package[92]. To control for multiple comparisons, p-values were adjusted using the Benjamini-Hochberg (BH) method to limit the False Discovery Rate (FDR). To ensure robust and reliable results, offspring counts that were abnormally low (below five) or high (above 55) over a three-day egg-laying period were excluded from the analysis. As a result, four families were removed.

*Development.* Developmental time analysis was performed using the Cox proportional hazards model[93], with the coxph function from the survival package[94], to assess the effect of phenotype on pupation and adult emergence times (Extended Data Fig. 4d).

*Mendelian inheritance.* A Chi-square goodness-of-fit test (stats package) was performed to assess whether the observed *reduced-eye* proportions in the F2 generation conformed to the expected Mendelian 3:1 segregation pattern (75% *normal-eye*, 25% *reduced-eye*, Fig. 2b). Observed and expected proportions for '*normal-eye*' and '*reduced-eye*' phenotypes were calculated for each treatment using the dplyr package[89]. To ensure robust statistical power, nine families that produced fewer than 10 offspring were excluded. The average family size after exclusion was 27.43, with a range of 45–58.

*Environmental influence on the expression of reduced-eye phenotype.* The environmental influence on the expression of the *reduced-eye* phenotype was analyzed in offspring produced from *normal-eye* parents maintained under different light conditions (standard light/dark cycle and continuous light, Fig. 2c). The trait penetrance was evaluated by creating a binomial response variable, combining the counts of *reduced-eye* (as the number of successes) and *normal-eye* phenotypes (as the number of failures). A generalized linear model (GLM) from the stats package was fitted with light condition as the predictor, using a binomial distribution with a logit link function.

*Effect of the HDAC inhibitors NaB and TSA on the reversion of reduced-eye phenotype.* The effect of NaB and TSA on the *reduced-eye* phenotype (Extended Data Fig. 6) was evaluated in the same manner as above, using the GLM with a binomial error distribution and a logit link function. Offspring from families treated with NaB and TSA were analyzed to assess their effect on the *reduced-eye* phenotype. Families producing fewer than 10 offspring were excluded from the analysis (NaB experiment: 9 families excluded; NaB and TSA: 17 families excluded). The average family size was 22.8 offspring (range: 4-10) for NaB and 21.1 offspring (range: 8-15) for NaB and TSA experiment. The treatment effects were evaluated by comparing the occurrence of *normal-* versus *reduced-eye* offspring, with treatment type as the predictor.

### Reporting summary
Further information on research design is available in the Nature Portfolio Reporting Summary linked to this article.

## Data availability

All data supporting the findings of this study are available in the Supplementary Information file. All raw data generated in this study are provided in the Source Data files deposited on Zenodo (https://doi.org/10.5281/zenodo.17120480). The raw sequencing data have been submitted to the NCBI Sequence Read Archive SRA.

## Code availability

All custom scripts used to generate figures and analyze data in this study (including R scripts) are deposited on Zenodo and are available at (https://doi.org/10.5281/zenodo.17120480).

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

## Acknowledgements

We would like to acknowledge Prof. Dr J.G. and Prof. Dr T.F. for their valuable comments and support. We thank Barbara Hasert, Kathrin Brüggemann, Hildegard Schwitte and Una Hadziomerovic for their technical assistance. Sina Flügge for providing drawings of pupae and adult beetle used in the experimental design diagrams in Fig. 1a and Extended Data Fig. 3a–c. We would like to thank the Deutsche Forschungsgemeinschaft (DFG, German Research Foundation) for funding this work through the Research Training Group GRK 2220 "Evolutionary Processes in Adaptation and Disease" (project number 281125614 to JK) and as part of the SFB TRR 212 (NC³)–Project numbers 316099922 and 396780003 (to JK). Also, thanks to the German Academic Exchange Service (DAAD) for their financial support through PhD scholarships awarded to RS and ÖS.

## Author contributions

R.S. and J.K. conceptualized the project. R.S., Ö.S., R.R., N.K.E.S., R.P. and J.K. designed the experiments. R.S., Ö.S., R.R. and T.P. performed the lab experiments. M.E. and L.S. performed the bioinformatic analysis. R.S., Ö.S., R.R. and N.K.E.S. performed the statistical analyses. R.S. wrote the original manuscript draft with support from J.K. and L.S. L.S. and N.K.E.S. contributed equally to this work. All authors commented and revised the manuscript and approved the final version for publishing.

## Funding

## Competing interests

The authors declare no competing interests.
