## [Peer Review file · Nature Communications]

HSP90 as an Evolutionary Capacitor Drives Adaptive Eye Size Reduction via Atonal

Corresponding Author: Professor Joachim Kurtz

Version 0:

Reviewer comments:

Reviewer #1

(Remarks to the Author)

Overview:

I was excited to read this manuscript because the abstract suggested a significant advance and proof-of-principle for heritable, adaptive uncovering of cryptic genetic variation via Hsp90 inhibition. By and large, I think this promise from the abstract was met by the study, I did very much enjoy reading the manuscript, and I think it has a lot to add to the field. However, there are several points that need to be clarified, potentially with additional data, for the manuscript to have an even more compelling story.

Major Comment:

The major shortcoming I see (that potentially extends to other studies of Hsp90 and CGV) is that we are not given evidence that Hsp90 expression or availability is affected by environmental change/stress. The authors functionally manipulate Hsp90 with RNAi and drugs, then assay fitness in different environments. We don't know however, if the different environment--continuous light--affects Hsp90. I think the most compelling case that corresponds to that laid out in this study would require:

- 1) Establishing a link between environment and Hsp90 expression/availability
- 2) Establishing a link between environment and phenotypic variation
- 3) Establishing a link between Hsp90 expression and phenotypic variation
- 4) Establishing link between phenotypic variation and fitness, especially in the inducing environment
- 5) Establishing that phenotypic variation is inherited across generation

The authors address points 3-4, but 1 & 2 remain untested, perhaps in any system. On L72-74 for instance, it is unclear if the references refer to the first clause "However, HSP90 availability may become limited under stressful environmental conditions," or the second clause, "as it is required by numerous proteins that become damaged during stress" or both. If there are no references to support the first clause, then it seems like a major gap that seeks filling.

The authors should consider the effects of constant light itself of phenotype production and HSP90 expression. They ought to consider discussing how, in nature for example, a reduction of HSP90 expression (akin to their manipulations) could occur and have fitness differences be borne out in a novel environment (e.g. constant light) if that environment does or does not affect (we don't know from the current data) the expression itself.

Additional comments:

L43: note what kind of variation is meant here

L102-106: The reference to the previous study on social cues makes the reader think that this will be an important element of the current study even though it is not. The authors should consider ending with a brief overview of their study, its phenotypes, environments, and aims.

L108: "HSP90 reduction, mimicking what may occur during environmental stress, in ..." Again, evidence of HSP90 with environmental stress seems like a key point that is not well-established from what I see in the manuscript.

L119-122 seems to disagree with the figure 1 legend. The figure says F2 had 27.8%

L138 Why not Hsp83? It seems odd that a different gene is being assayed here when everything else focused on Hsp83.

L158: How were reduced- and normal-eye beetles operationally delimited? Knowing the cut-off metric is important because much of the phenotypic and fitness data between the groups overlap (Fig. 1).

L210 Please note that the deviation was lower than expected.

L273-274 I think omission of some commas in the sentence make it a bit awkwardly worded. Do you mean "Notably, our mapping analysis shows that the atonal gene, which consists of a single exon, overlaps with four synonymous polymorphic SNPs across the four pooled samples.."?

L270 & 273 Consider removing "Notably" from one of these sentences.

L279 I'm sure the authors wish they could pinpoint more closely candidate regulatory elements. One possibility is to bioinformatically identify differences in TF motifs between lines in the candidate region and how the presence/absence of SNPs affect motif occurrences. The online resource MEME Suite might be useful to explore and get a bit closer to the actual causal SNP(s). <https://meme-suite.org/meme>. Of course, CHIP-seq or ATAC-seq could also identify some candidate positions.

L373 This is the closest we get to understanding if HSP90 expression is affected by stress. It is unfortunate that it is not in the environment used in this study.

L601: I am confused about the abnormal leg phenotype and its being reported. Did this always co-occur with the reduced eyes? If not, how often was reduced-eyes by itself and why weren't these lines chosen instead of mixed phenotype lines? Why were leg phenotypes used to establish eye phenotype lines?

L641: Again, unclear why Hsp83 wasn't measured.

L680-681: This is a clever design.

Fig. 1

I don't understand the diagram in 1a, what are the small arrows and red legs highlighting? It doesn't say in the legend. In 1b, % frequency for the y-axis is odd. I'm not sure how to interpret it. The legend says that the % eye ptype is above the gold bar, but those numbers are more likely sample sizes. I think something must be missing. In 1d, how were groups categorized, especially since there is overlap with some normal-eye having areas smaller than reduced-eye? Should it be treatment or line rather than ptype for the colors in panels in d-f?

Reviewer #2

(Remarks to the Author)

The study by Sayed et al. explores the underappreciated evolutionary role of Hsp90 buffering in animals, identifying a beetle eye size trait whose variation is masked by Hsp90. Their findings suggest that Hsp90 enables the accumulation of cryptic genetic variation in natural populations. Transient Hsp90 inhibition (via chemical or siRNA) reveals a reduced-eye phenotype that persists in a subset of F1 and F2 progeny, can be driven to near fixation through crosses, and shows fitness advantages under constant light. A dimorphism is noted, and atonal is proposed as a phenocopy for Hsp90 inhibition. These findings are important and intriguing.

While this is a compelling example of Hsp90 buffering in animals, the manuscript suffers from overstated claims, limited novelty, and interpretational issues. The conclusions need to be tempered or supported with additional data. The introduction and discussion should better contextualize the work within existing evidence supporting the adaptive value of Hsp90-buffered variation, avoiding claims of controversy or primacy. The experimental rigor and statistical analyses also require improvement. That said, the manuscript is clearly written and presents an intriguing story. I would support publication following major revisions to address the concerns outlined below.

Major issues:

1. Conceptual Framing and Literature Context: The manuscript overstates the novelty and conceptual primacy of Hsp90 as an evolutionary capacitor. While this study advances our understanding of Hsp90-buffering in animals, the concept is well-established, particularly in yeast. The authors misrepresent the state of the field by framing the adaptive value of Hsp90-buffered variation as controversial. Multiple prior studies, including those by Lindquist and others (e.g., PMID: 16195452, 21205668, 39052788), have clearly demonstrated adaptive, genetically defined Hsp90-buffered traits. The manuscript also lacks appropriate citations and introduces misleading statements about the novelty and controversy of this mechanism. All novelty claims should be carefully rephrased to reflect that the contribution refers to Hsp90's role in animal evolution not evolution in general. I suggest revising lines 27-31, 37-38, 78-79, 86-87, 98, 169-170, 283-284, and 357-358 (these may not be all the sentences in need of revision to address this issue).

2. The ecological and adaptive relevance of the reduced-eye trait is inadequately supported. The authors test it under artificial, non-ecological conditions (e.g., continuous light), without exploring whether it is beneficial or deleterious under more natural environments. As a loss-of-function phenotype, reduced-eye is unlikely to confer broad fitness advantages, and its frequency in wild populations remains unexamined. Strong claims about its evolutionary significance require broader validation and testing of alternative hypotheses, including potential fitness costs. Overall, the trait's ecological relevance is unclear, and the study lacks the rigor seen in comparable work on Hsp90-buffered traits in yeast.

3. The claim that atonal mediates Hsp90-buffered eye-size variation is not sufficiently demonstrated. While atonal RNAi phenocopies the trait, there is no genetic or expression evidence linking atonal to Hsp90 buffering in the beetle. Allele-specific expression, response to Hsp90 inhibition, and functional SNP data are lacking. The role of HDACs is weakly supported and lacks a positive control. Without stronger evidence, the proposed mechanism remains speculative and should be presented as such.

4. Several conclusions overreach the presented data. Key findings (e.g., persistence of the trait, its selection, and its mechanistic basis) have low penetrance, limited replicates, and insufficient statistical support. Alternative hypotheses are not tested, and the environmental relevance of findings is unclear. Overall, stronger controls, clearer definitions, and more cautious interpretation are needed throughout. Statistical analyses are not robust. Effects on fitness are weak and may not be significant after multiple hypothesis testing.

Minor comments:

1. In place of "Hsp90-released trait" I suggest "Hsp90-buffered trait"; lines 38 and 74.

2. The keyword "plasticity" was not mentioned. Remove keyword or add explanation of the term in the text.

3. Statistical significance, * should be replaced with increments **, ***, ****. A single asterisk (*) indicating a p-value less than 0.05 (significant), two asterisks (**) a p-value less than 0.01 (highly significant), and three asterisks (***) indicate a p-value less than 0.001 (very highly significant).

4. Hsp90 is an epistatic regulator but no epistasis experiments were performed.

5. Abstract lines 33-34: "stably inherited without continued HSP90 disruption" implies all revealed variation was stable but the data show closer to 20% of the revealed variation is inherited and further declines in each generation so it is not that stable.

6. Figures should be revised to improve clarity and intuitiveness. In particular, the presentation of penetrance—such as in the HDAC inhibition example—is often unclear and could benefit from more straightforward visualization. A more quantitative trait would be the number of ommatidia per animal, which although labor intensive seems to present with a great dynamic range and could foster a more robust statistical analysis approach. I understand that counting the number of ommatidia in each animal may not be feasible, so this is only a minor comment/suggestion.

Reviewer #3

(Remarks to the Author)

Version 1:

Reviewer comments:

Reviewer #1

(Remarks to the Author)

NCOMMS-25-2699A

My biggest concerns in the previous version of the manuscript were the links between Hsp90 levels, phenotypic variation, and environmental conditions. In their response letter, the authors do a convincing job of explaining that the conditions that affected Hsp90 availability need not be the same conditions that select upon phenotypes and genotypes revealed by Hsp90 depletion. Further, they explained the link between measuring expression level of Hsp68 instead of Hsp83. By and large, I am satisfied with their responses to my initial comments and suggestions, and I am pleased by their inclusion of additional data and consideration of my comments. I have only a few suggestions for further clarifying these points in the text.

L78: Consider mentioning here that selection could be acting in the Hsp90 affecting environment or a different environment. "...differences on which selection can act either in the disrupting environmental conditions or other conditions altogether."

L186 consider adding a reminder here that continuous light COULD be encountered by these beetles in their human-commensal environment

L208 Consider changing “certain” to “some, albeit unknown, frequency”

L321 “...unmasking cryptic genetic variation and producing new phenotypes.” To be explicit about the consequence of that CGV being uncovered.

(Remarks on code availability)

Reviewer #2

(Remarks to the Author)

All my comments have been fully addressed. I congratulate the authors on their beautiful work and approve of its publication.

(Remarks on code availability)

RESPONSE TO THE REVIEWER COMMENTS

Dear Reviewers,

We appreciate the opportunity to revise our manuscript entitled “HSP90 as an Evolutionary Capacitor Drives Adaptive Eye Size Reduction via *atonal*.” We are grateful for the reviewers’ insightful comments and suggestions, which have helped us to improve the clarity and quality of our work.

Reviewer #1 (Remarks to the Author):

Overview:

I was excited to read this manuscript because the abstract suggested a significant advance and proof-of-principle for heritable, adaptive uncovering of cryptic genetic variation via Hsp90 inhibition. By and large, I think this promise from the abstract was met by the study, I did very much enjoy reading the manuscript, and I think it has a lot to add to the field. However, there are several points that need to be clarified, potentially with additional data, for the manuscript to have an even more compelling story.

response:

Thank you very much for your positive feedback and for recognizing the strengths and significance of our study. We are pleased that you found the manuscript both promising and enjoyable, and we appreciate your suggestions for clarifications and additional data to further strengthen the work. As detailed in the following, we added additional data as suggested.

Major Comment:

The major shortcoming I see (that potentially extends to other studies of Hsp90 and CGV) is that we are not given evidence that Hsp90 expression or availability is affected by environmental change/stress. The authors functionally manipulate Hsp90 with RNAi and drugs, then assay fitness in different environments. We don’t know however, if the different environment---continuous light---affects Hsp90. I think the most compelling case that corresponds to that laid out in this study would require:

- 1) Establishing a link between environment and Hsp90 expression/availability
- 2) Establishing a link between environment and phenotypic variation
- 3) Establishing a link between Hsp90 expression and phenotypic variation
- 4) Establishing link between phenotypic variation and fitness, especially in the inducing environment
- 5) Establishing that phenotypic variation is inherited across generation

The authors address points 3-4, but 1 & 2 remain untested, perhaps in any system. On L72-74 for instance, it is unclear if the references refer to the first clause “However, HSP90 availability may become limited under stressful environmental conditions,” or the second clause, “as it is required by numerous proteins that become damaged during stress” or both. If there are no references to support the first clause, then it seems like a major gap that seeks filling.

response:

Thank you for this important and insightful comment. We have conducted additional experiments to further substantiate the link between HSP90 and the *reduced-eye* phenotype, now showing that RNAi knockdown of *Hsp83* reduces *atonal* expression, supporting a regulatory link (Fig.3g) and that *atonal* is consistently downregulated in *reduced-eye* monomorphic lines (Fig.3f). More details on these data are provided in our response to Reviewer 2. While these additional data provide further support for point 3), we also carefully considered points 1) and 2) and conducted an additional experiment to address these points (see below). While agreeing that directly linking environmental changes to HSP90 expression or availability could strengthen both our manuscript and the broader understanding of HSP90's role in adaptation, we would like to provide some conceptual clarifications.

Our study is conceptually grounded in the mechanism whereby HSP90 becomes limiting primarily under physiological stresses, in particular those that cause protein damage or denaturation – such as thermal, toxic, pathogenic or osmotic stress or other proteotoxic conditions – where demand for HSP90 is elevated and its availability for other cellular processes becomes restricted. Under these conditions, HSP90 is required for the stabilization and refolding of numerous damaged proteins, which may temporarily reduce its availability and thereby unmask cryptic genetic variation. We would like to point out that according to this concept, the environment leading to reduced HSP90 availability is not necessarily the same as the environment in which released phenotypes may show fitness advantages. Specifically, the environments in points 1) and 2) are not expected to be the same as the environments where fitness benefits manifest (point 4).

We also acknowledge your point regarding the need for a clear reference supporting this mechanism. In response, we identified and now cite Alford and Brandman (2018), which directly supports the statement that HSP90 availability can become limited under stressful conditions that challenge proteostasis. We have revised the relevant sentence in the manuscript for clarity and included this reference at line 83-86. The revised sentence now reads:

“However, under stressful environmental conditions that disrupt proteostasis, HSP90 availability may become limited due to its involvement in stabilizing numerous proteins that are damaged or denatured during stress (Alford and Brandman 2018; Borkovich et al. 1989; Chen and Wagner 2012; Peuss et al. 2015)”.

The authors should consider the effects of constant light itself of phenotype production and HSP90 expression. They ought to consider discussing how, in nature for example, a reduction of HSP90 expression (akin to their manipulations) could occur and have fitness differences be borne out in a novel environment (e.g. constant light) if that environment does or does not affect (we don't know from the current data) the expression itself.

response:

Thank you for raising this point regarding the potential effects of constant light on both phenotype production and HSP90 expression, as well as the ecological context for HSP90 reduction and fitness in novel environments. However, as noted above, please consider that it is conceptually not required that it is the same environmental condition that leads to reduced HSP90 availability and the phenotypes' fitness advantage.

- To nevertheless address your comment, we conducted an experiment to directly test whether constant light affects HSP90 expression. Our results showed no significant change in HSP90 expression under constant light, supporting our expectation that this condition alone does not challenge proteostasis or induce a heat shock response. We did not include these additional data into the manuscript but are happy to provide them if requested.
- Regarding the production of the *reduced-eye* phenotype, our data indicate that the phenotype is only revealed when HSP90 function is experimentally reduced (via RNAi or chemical inhibition), not by constant light alone. This suggests that constant light acts as a selective environment, but not directly as an inducer of the phenotype.
- In nature, HSP90 limitation is most likely to occur under environmental stresses that cause protein damage or denaturation, where HSP90 becomes limited due to increased demand. In such scenarios, hidden genetic variation could be revealed, and, as our study demonstrates, may confer a fitness advantage in specific novel environments like continuous light.
- We have expanded the discussion in the revised manuscript to clarify these points and to address how natural environmental stresses could lead to HSP90 limitation and the expression of adaptive phenotypes, even if the novel environment itself (e.g., constant light) does not directly affect HSP90 expression. We have added the following sentences at line 368 in the Discussion:

“It is well established that certain environmental stresses can limit HSP90 availability by increasing the demand for its chaperone function (Alford and Brandman 2018), thereby unmasking cryptic genetic variation (Rutherford and Lindquist 1998). Our findings suggest that once such variation is exposed—regardless of the initial stressor—it can become subject to selection in a novel environment, even if that environment does not directly affect HSP90 expression.”

Additional comments:

L43: note what kind of variation is meant here

response:

We clarified the type of variation in the revised manuscript. The sentence now reads (line 51):

*“Canalization buffers development against genetic and environmental disturbances, producing stable phenotypes despite underlying **genetic and environmental** variation (Waddington 1942)”.*

L102-106: The reference to the previous study on social cues makes the reader think that this will be an important element of the current study even though it is not. The authors should consider ending with a brief overview of their study, its phenotypes, environments, and aims.

response:

Thank you for pointing this out. To improve clarity and better guide the reader into our study, we have revised the final paragraph of the Introduction as suggested and now include a

concise overview of our study's main phenotypes, environments, and aims. This ensures a clear transition into the Results section without implying undue emphasis on social cues. Now this section reads _line 118:

*“Here we used the red flour beetle, *Tribolium castaneum* as an alternative insect model (Brown et al. 2009; Pointer, Gage, and Spurgin 2021) to examine the genetic basis and fitness relevance of an HSP90-released phenotype. A previous study in *T. castaneum* found consistent downregulation of *Hsp83*, the primary HSP90-coding gene, in response to social cues mimicking a stressful environment. This suggests that HSP90 levels may be adaptively regulated, potentially facilitating the release of cryptic genetic variation when advantageous (Peuss et al. 2015). **In the present study, we directly manipulated HSP90 function in *T. castaneum* and characterized the resulting reduced-eye phenotype. We identified the gene causing this phenotype and assessed its fitness consequences to evaluate the adaptive potential of HSP90-released variation.**”*

L108: “HSP90 reduction, mimicking what may occur during environmental stress, in ...”
Again, evidence of HSP90 with environmental stress seems like a key point that is not well-established from what I see in the manuscript.

response:

Following your suggestion, we have revised the sentence at line 136 for greater clarity. As mentioned above, we have also addressed the evidence for HSP90 limitation under environmental stress and included the relevant reference (Alford and Brandman 2018) in the manuscript. The sentence now reads:

*“To investigate the effects of reduced HSP90 function—mimicking conditions that may arise during environmental stress—we used RNA interference (RNAi) to target *Hsp83* in *T. castaneum*, with the aim of releasing HSP90-buffered, selectable phenotypic variants in *Cro1*, a genetically heterogenous wildtype population of this beetle”*

L119-122 seems to disagree with the figure 1 legend. The figure says F2 had 27.8%

response:

We appreciate your careful reading. The percentage shown in the original figure 1a (27.8%) represented the combined incidence of the *reduced-eye* phenotype in F2 offspring from the two families in which the trait was observed (12/47 and 20/68, totaling 32/115 beetles). To avoid confusion, we have now removed this percentage from Figure 1a and present the detailed incidence rates only in the text and Extended Data Table 1. The text now clearly states the incidence rates for each family (25.5% and 29.4%, respectively), as well as the overall frequency (4.2%) among all F2 offspring screened within this treatment as shown in Fig. 1b. This should resolve any inconsistency between the figure and the text.

L138 Why not *Hsp83*? It seems odd that a different gene is being assayed here when everything else focused on *Hsp83*.

response:

We appreciate this observation. We focused on *Hsp68a* (HSP70 family) as a commonly used readout for HSP90 limitation, because 17-DMAG inhibits HSP90 (encoded by *Hsp83*) at the protein level rather than by reducing its mRNA expression. 17-DMAG is a well-characterized HSP90 inhibitor that binds to the ATP-binding pocket of the HSP90 protein, thereby blocking

its chaperone activity and leading to functional inhibition (Jez et al. 2003; Trepel et al. 2010). This inhibition does not typically reduce *Hsp83* mRNA levels. Instead, inhibition of HSP90 activity is known to induce a compensatory heat shock response, including upregulation of HSP70 family genes such as *Hsp68a* (Kudryavtsev et al. 2017; Zhou et al. 2013). Thus, increased *Hsp68a* expression serves as a reliable molecular marker of successful HSP90 inhibition in our system. To clarify this, we have added the following sentence to the manuscript_line 181:

“17-DMAG inhibits HSP90 at the protein level by binding to its ATP-binding pocket, thereby blocking its chaperone activity without affecting Hsp83 mRNA levels (Jez et al. 2003; Trepel et al. 2010). As a result, upregulation of HSP70 family genes, such as Hsp68a, serves as a molecular marker of successful HSP90 inhibition(Jez et al. 2003).”

L158: How were reduced- and normal-eye beetles operationally delimited? Knowing the cut-off metric is important because much of the phenotypic and fitness data between the groups overlap (Fig. 1).

response:

Thank you for pointing this out, it is an important point that needed clarification. During development, *reduced-* and *normal-eye* beetles could not be distinguished at the larval or pupal stages based on morphology alone. However, because individuals were tracked throughout development from larvae to adults, their adult eye phenotype could be retrospectively assigned to earlier stages. In the adult stage, classification into “*reduced-eye*” and “*normal-eye*” was based on visual inspection under a stereomicroscope, supported by quantitative image analysis of eye area. *Reduced-eye* beetles consistently showed visibly smaller eyes. For quantification, we normalized eye area within each developmental stage and used the adult phenotype (clearly visible) to assign individuals into the two groups. We have clarified this in the figure legend Fig1d as follows: line 1120

“Fig.1.....d, Normalized eye area in normal-eye and reduced-eye individuals across the three developmental stages, showing significant differences in eye size within stage. Normalization was done by stage's mean. Lines connect individual beetles, which were tracked from larval to adult stage. Phenotype classification (normal- vs. reduced-eye) was based on adult morphology, which allowed retrospective assignment of phenotype to earlier developmental stages.”

And this sentence was added to the result part: line 205

“Because individuals were tracked throughout development, their adult phenotype could be retrospectively assigned to their larval and pupal stages.”

L210 Please note that the deviation was lower than expected.

response:

Indeed, the deviation was in the direction of a lower-than-expected frequency of the reduced-eye phenotype. We have clarified the sentence in the manuscript to reflect this more precisely. The updated sentence now reads: line 261

“However, in the crosses RNAi ♀ - Cro1 ♂ and RNAi ♂ - Cro1 ♀, we observed a significantly lower proportion of the reduced-eye phenotype than the expected 25 %...”

L273-274 I think omission of some commas in the sentence make it a bit awkwardly worded. Do you mean “Notably, our mapping analysis shows that the atonal gene, which consists of a single exon, overlaps with four synonymous polymorphic SNPs across the four pooled samples..”?

response:

We have revised the sentence in L336 for improved clarity. The revised sentence now reads:

“Our mapping analysis shows that the atonal gene—which consists of a single exon—overlaps with four synonymous polymorphic SNPs across the four pooled samples”

L270 & 273 Consider removing “Notably” from one of these sentences.

response:

Done! Line 337 the word “Notably” is removed.

L279 I’m sure the authors wish they could pinpoint more closely candidate regulatory elements. One possibility is to bioinformatically identify differences in TF motifs between lines in the candidate region and how the presence/absence of SNPs affect motif occurrences. The online resource MEME Suite might be useful to explore and get a bit closer to the actual causal SNP(s). <https://meme-suite.org/meme>. Of course, CHIP-seq or ATAC-seq could also identify some candidate positions.

response:

We thank you for this suggestion. Following the recommendation, we performed motif discovery using the MEME Suite to identify potential transcription factor binding motifs in the candidate region. We also explored publicly available ATAC-seq and FAIRE-seq datasets from the iBeetleBase genome browser (<https://ibeetle-base.uni-goettingen.de/genomebrowser/>) to examine chromatin accessibility.

While the MEME analysis identified transcription factor motifs overlapping several genes in our candidate region, including atonal, the ATAC-seq and FAIRE-seq data revealed strong peaks of chromatin accessibility within the region lacking annotated genes but enriched for highly differentiated SNPs. These results support the hypothesis that a cis-regulatory element in this intergenic region may be influencing atonal expression. We have added these findings to the manuscript and included a new figure illustrating both the motif locations and peaks of chromatin accessibility. We now write in L341:

“This suggests that the phenotype may instead be caused by an as-yet-undiscovered regulatory sequence regulating atonal expression. We used two publicly available ATAC-seq (Mau et al. 2023) and FAIRE-seq (Lai et al. 2018) datasets to explore potential cis-regulatory regions. Particularly an area lacking annotated genes, yet enriched in highly differentiated SNPs, showed peaks of chromatin accessibility in both ATAC-seq and FAIRE-seq datasets (Extended Data Fig. 9).”

Moreover, we have added the following description to the Methods section detailing how MEME was ran_ line: 954

“To identify potential regulatory motifs within the candidate region, we used the MEME Suite (Bailey and Elkan 1994) (<https://meme-suite.org>) to perform de novo motif discovery. We used the full 100 kb candidate region as input and ran the analysis with default parameters.”

L373 This is the closest we get to understanding if HSP90 expression is affected by stress. It is unfortunate that it is not in the environment used in this study.

response:

We recognize the importance of directly linking HSP90 regulation to specific environmental conditions. However, as mentioned before, it is conceptually not required that it is the same environmental condition that leads to reduced HSP90 availability and the phenotypes' fitness advantage. Our earlier work in *T. castaneum* demonstrated downregulation of HSP90 transcripts (*Hsp83*) in response to stressful social cues, specifically the presence of wounded conspecifics (Peuss et al. 2015). Other studies, in both *T. castaneum* and various insects, show that HSP90 expression is modulated by multiple forms of environmental stress, such as temperature extremes and chemical exposure (Brom et al. 2015; Ding et al. 2021; Liang et al. 2023). However, as you correctly note, our present study did not measure HSP90 levels under the exact selection regimes implemented here. We have now addressed this point in the revised discussion and have more clearly stated the need for future work that directly investigates HSP90 regulation in the context of these environments.

Line 469

“In a previous study in *T. castaneum*, HSP90 downregulation was observed in a risky environment, i.e., in cohabitation with wounded conspecifics (Peuss et al. 2015). Although this regulatory response was not directly addressed in the present study, the two studies together suggest a scenario that links evolutionary capacitance—and thus evolvability of a population—to environmental conditions...”

L601: I am confused about the abnormal leg phenotype and its being reported. Did this always co-occur with the reduced eyes? If not, how often was reduced-eyes by itself and why weren't these lines chosen instead of mixed phenotype lines? Why were leg phenotypes used to establish eye phenotype lines?

response:

We appreciate the opportunity to clarify this point. Initially, F1 beetles exhibiting visible morphological abnormalities – most commonly leg malformations – were selected for generating F2 lines. This approach was primarily practical: such abnormalities served as visible markers of developmental disruption, suggesting that cryptic genetic variation might have been unmasked. The goal was to increase the chance of observing heritable phenotypes such as the *reduced-eye* trait in the F2 generation.

When the *reduced-eye* phenotype was first observed in F2 offspring from abnormal-leg F1 beetles, we tested whether this trait was linked to the leg malformation by also crossing their normal-legged full siblings from the same RNAi parental families. These crosses also yielded reduced-eye offspring in the F2, indicating that the *reduced-eye* and leg malformation phenotypes were not genetically linked.

It is also important to clarify that our *reduced-eye* lines were established only after observing this specific phenotype in the F2 generation. While these were initially derived

from families in which the F1 showed leg abnormalities, the *reduced-eye* lines themselves did not later display leg malformations, and the eye phenotype became monomorphic in subsequent generations.

To reflect this in the manuscript, we have added the following clarification in the Materials and Methods section: line 725

“... Notably, the *reduced-eye* phenotype was not genetically linked to leg malformations, as it also appeared in F2 offspring from normal-leg F1 siblings of the same RNAi-treated families (unpublished data).”

L641: Again, unclear why Hsp83 wasn't measured.

response:

This point is now clarified in the Results section, where we explain that 17-DMAG inhibits HSP90 (encoded by *Hsp83*) at the protein level, not at the level of mRNA expression. Therefore, *Hsp83* mRNA is not expected to respond directly to inhibition.

L680-681: This is a clever design.

response:

Thank you – we appreciate the positive feedback on our experimental design.

Fig. 1

I don't understand the diagram in 1a, what are the small arrows and red legs highlighting? It doesn't say in the legend.

In 1b, % frequency for the y-axis is odd. I'm not sure how to interpret it. The legend says that the % eye ptype is above the gold bar, but those numbers are more likely sample sizes. I think something must be missing.

In 1d, how were groups categorized, especially since there is overlap with some normal-eye having areas smaller than reduced-eye? Should it be treatment or line rather than ptype for the colors in panels in d-f?

response:

Thank you for your careful evaluation and helpful observations regarding Figure 1. We address each point below:

1. **Figure 1a – red arrows and legs:** You are absolutely right – the small arrows were intended to indicate that the F2 *reduced-eye* phenotypes originated from F1 beetles with abnormal leg phenotypes (highlighted in red). These represented the families from which *reduced-eye* lines were established. However, as clarified in the text, we later confirmed that the *reduced-eye* phenotype is not genetically linked to leg malformations, since it also appeared in F2 offspring from normal-legged siblings of the same F1 families. To avoid confusion, we have removed these small arrows and red leg highlights from the updated version of the figure.
2. **Figure 1b – % frequency and legend labeling:** It is correct that the y-axis represents the **frequency (%) of the *reduced-eye* phenotype** within the screened F2 beetles in each treatment. The values **above the bars** correspond to **sample sizes**, not phenotypic percentages. The note in the legend stating that “% eye ptype is above

the gold bar” was left over from an earlier version of the figure and was mistakenly not removed when the plot format was updated. We have now corrected this in the revised legend.

3. **Figure 1d – group categorization and classification:** As noted earlier in this response letter and the manuscript, normalization of eye size was performed within each developmental stage. Individual beetles were tracked throughout development, and adult eye phenotype (normal or reduced) was assigned based on adult morphology using clear visual criteria, allowing retrospective classification across earlier stages. While there is some overlap in eye area between phenotypic classes, classification was based on **adult appearance**, which was visually distinct and confirmed by additional morphological analyses (see Fig. 1e for ommatidia counts). We’ve clarified this further in the legend to avoid ambiguity.
4. **Color coding in panels d–f:** We used eye phenotype (normal vs. reduced) as the common variable to categorize data in panels d–f to ensure consistency across the subfigures. Treatments or source lines varied across panels and were not consistently applicable, while eye phenotype provided a meaningful and common dimension for comparison. We note that:
 - **Panel d** compares normalized eye area by phenotype across developmental stages
 - **Panel e** compares ommatidia counts by adult eye phenotypes
 - **Panel f** compares reproductive output by phenotype under two environmental conditions

Reviewer #2 (Remarks to the Author):

The study by Sayed et al. explores the underappreciated evolutionary role of Hsp90 buffering in animals, identifying a beetle eye size trait whose variation is masked by Hsp90. Their findings suggest that Hsp90 enables the accumulation of cryptic genetic variation in natural populations. Transient Hsp90 inhibition (via chemical or siRNA) reveals a reduced-eye phenotype that persists in a subset of F1 and F2 progeny, can be driven to near fixation through crosses, and shows fitness advantages under constant light. A dimorphism is noted, and *atonal* is proposed as a phenocopy for Hsp90 inhibition. These findings are important and intriguing.

response:

Thank you very much for your positive and thoughtful feedback on our work. We're glad that you found the conceptual approach and the main findings important and intriguing. Your encouraging comments were very motivating during the revision process and helped us further refine the manuscript.

While this is a compelling example of Hsp90 buffering in animals, the manuscript suffers from overstated claims, limited novelty, and interpretational issues. The conclusions need to be tempered or supported with additional data. The introduction and discussion should better contextualize the work within existing evidence supporting the adaptive value of Hsp90-buffered variation, avoiding claims of controversy or primacy. The experimental rigor and statistical analyses also require improvement. That said, the manuscript is clearly written and presents an intriguing story. I would support publication following major revisions to address the concerns outlined below.

response:

Thank you for the thoughtful and constructive review. We have made substantial revisions to address your concerns:

- **Tempered claims and improved context:** We revised the Abstract, Introduction, and Discussion to avoid overstating novelty or controversy. The study is now clearly framed as an extension of previous work, highlighting HSP90's buffering in an animal under a certain environmental condition. We now more clearly acknowledge prior evidence for Hsp90 buffering and clarify where the novelty of our findings lies.
- **New supporting data:**
 - RNAi knockdown of *Hsp83* reduces atonal expression, supporting a regulatory link (Fig.3g).
 - Atonal is consistently downregulated (~20%) in *reduced-eye* monomorphic lines (Fig.3f).
 - Computational analyses (Extended Data Fig. 9) reveal conserved cis-regulatory motifs within accessible chromatin near candidate loci.

These additions strengthen the mechanistic basis of our conclusions and address interpretational concerns. We appreciate your feedback and believe the manuscript is now significantly improved

Major issues:

1. Conceptual Framing and Literature Context: The manuscript overstates the novelty and conceptual primacy of Hsp90 as an evolutionary capacitor. While this study advances our understanding of Hsp90-buffering in animals, the concept is well-established, particularly in yeast. The authors misrepresent the state of the field by framing the adaptive value of Hsp90-buffered variation as controversial. Multiple prior studies, including those by Lindquist and others (e.g., PMID: 16195452, 21205668, 39052788), have clearly demonstrated adaptive, genetically defined Hsp90-buffered traits. The manuscript also lacks appropriate citations and introduces misleading statements about the novelty and controversy of this mechanism. All novelty claims should be carefully rephrased to reflect that the contribution refers to Hsp90's role in animal evolution not evolution in general. I suggest revising lines 27-31, 37-38, 78-79, 86-87, 98, 169-170, 283-284, and 357-358 (these may not be all the sentences in need of revision to address this issue).

response:

Tempering of claims and improved contextualization: We revised the Abstract, Introduction, and Discussion to avoid implying novelty or controversy where not appropriate. Statements suggesting primacy or debate (e.g., about the role of HSP90 buffering in animals) were rephrased to more clearly position our study within the broader literature. We agree that studies in yeast and other fungi have demonstrated adaptive, genetically defined HSP90-buffered traits, and make it now clearer that the novelty of our study lies in the fact that to our knowledge no study in any animal has so far connected a newly released, HSP90-buffered trait to its genetic basis and fitness consequences.

2. The ecological and adaptive relevance of the reduced-eye trait is inadequately supported. The authors test it under artificial, non-ecological conditions (e.g., continuous light), without exploring whether it is beneficial or deleterious under more natural

environments. As a loss-of-function phenotype, reduced-eye is unlikely to confer broad fitness advantages, and its frequency in wild populations remains unexamined. Strong claims about its evolutionary significance require broader validation and testing of alternative hypotheses, including potential fitness costs. Overall, the trait's ecological relevance is unclear, and the study lacks the rigor seen in comparable work on Hsp90-buffered traits in yeast.

response:

We thank you for highlighting these important points, which we believe raise valid considerations regarding the ecological context and evolutionary relevance. Below, we provide our response to each point individually.

- **Regarding the relevance of our experimental conditions:** While we acknowledge the concern regarding “artificial” or “non-ecological” laboratory conditions (e.g., continuous light), we respectfully note that *T. castaneum* is now a globally distributed species and a major pest of post-harvest grains and flour stored in **human-made environments** – including flour mills, warehouses, and storage bags – where beetles naturally encounter a range of light conditions (continuous, intermittent, or absent) and temperature fluctuations. Our laboratory setup, using flour as substrate and manipulating light and temperature, was specifically chosen to reflect these **ecologically relevant, real-world** habitats.

To further ensure relevance, we also evaluated fitness under **constant darkness** and **mild heat stress (+5 °C)** – conditions also common in grain storage environments – but found no apparent benefit of the reduced-eye phenotype under these treatments (data not shown). Therefore, the observed fitness advantage under continuous light appears context-dependent, but still grounded in environmental conditions *T. castaneum* naturally experiences in its current human-associated niche.

We believe this experimental design effectively mimics the abiotic factors and substrate conditions typical of modern *Tribolium* habitats, providing a meaningful context to assess the adaptive value of the reduced-eye trait. Pointer et al. (2021) stated that “Owing to *Tribolium*’s long human-commensal history (~70,000 generations), the laboratory medium also has the advantage of very closely approximating its semi-wild habitat in food-storage facilities, allowing a lab environment that is less abstracted than that in other insect models”.

- **Regarding the loss-of-function phenotype:** While we agree that the reduced-eye phenotype – likely representing a loss-of-function – may not provide broad or universal fitness advantages, accumulating evidence suggests that **loss or reduction of visual structures can be conditionally adaptive**, especially in low-light or energetically constrained environments. Several studies have proposed that eye reduction or loss may be favored evolutionarily due to significant energy savings (Jeffery 2009; Moran, Softley, and Warrant 2015; Niven 2015; Ri as-Sánchez et al. 2025). Visual processing, including maintenance of photoreceptors and neural tissue, is metabolically costly: for example, cave fish devote 5–17% of total metabolic energy to vision (Moran et al. 2015), and in *Drosophila melanogaster*, photoreceptor activity alone accounts for around 8% of the organism’s total energy usage (Laughlin, de Ruyter Van Steveninck, and Anderson 1998). Such costs rise in bright environments—energy demand may increase up to fourfold compared to darkness (Okawa et al. 2008).

Even minor reductions in eye size have been associated with decreased energetic expense through the positive scaling of visual neuropile and eye size (Niven 2015). Insects inhabiting dim or concealed microhabitats, such as the larval stages of *D. melanogaster* and *T. castaneum*, tend to exhibit behaviors and morphologies reflecting reduced reliance on vision (Busto, Iyengar, and Campos 1999; Park 1934). Since *T. castaneum* adults are also strongly photo-negative, reduced eye size could plausibly provide a selective advantage in the dark, resource-limited environments typical of grain storage habitats by minimizing the metabolic burden of maintaining unnecessary visual tissue.

- **Regarding the frequency of the phenotype in wild populations:** There is currently no published data on the frequency of the *reduced-eye* phenotype in wild or synanthropic populations of *T. castaneum*. While the trait has not been reported in field surveys, we believe it likely exists at very low frequency, given its spontaneous appearance in lab populations. Detecting it would require screening tens of thousands of individuals, which was beyond the scope of this study. Nonetheless, we agree this is an important and relevant direction for future research.

Finally, we recognize that the ecological relevance and evolutionary stability of this trait require further validation in field populations and across multiple environments, and we have revised the text to clarify that our conclusions concern **context-dependent fitness effects, not broad adaptive significance**. We also agree that testing for potential costs and evaluating trait frequency in natural populations remain important next steps.

2. The claim that *atonal* mediates Hsp90-buffered eye-size variation is not sufficiently demonstrated. While *atonal* RNAi phenocopies the trait, there is no genetic or expression evidence linking *atonal* to Hsp90 buffering in the beetle. Allele-specific expression, response to Hsp90 inhibition, and functional SNP data are lacking. The role of HDACs is weakly supported and lacks a positive control. Without stronger evidence, the proposed mechanism remains speculative and should be presented as such.

response:

Thanks for these constructive comments. We have performed additional experiments to address them:

- **Direct genetic and regulatory link:** To strengthen our interpretation that HSP90 affects expression of *atonal*, we conducted new experiments and found that RNAi knockdown of *Hsp83* significantly reduces *atonal* expression (new Fig. 3g), suggesting that *HSP90* positively regulates *atonal* transcript levels.
- **HDAC experiment and controls:** The HDAC inhibition experiment is preliminary and was designed to explore whether epigenetic mechanisms may underlie the buffering of *atonal* expression. Beetles were fed flour discs containing either HDAC inhibitors or solvent-only (ethanol) as a control; additionally, we included loose flour (our standard maintenance medium) as a negative control. Importantly, our new computational analyses (Extended Data Fig. 9) reveal conserved cis-regulatory motifs within accessible chromatin near candidate loci. We further tested *atonal* expression in our monomorphic reduced-eye lines and found it reduced compared to the ancestral Cro1 line (Fig. 3f). These results provide new evidence consistent with a regulatory link between chromatin state, *Hsp83* function, and *atonal* expression.
- **Caveats:** We acknowledge that allele-specific expression data, natural-variant analysis, and functional SNP validation are currently lacking. These are all valuable directions for future work, which we plan to take into account in the next phase of the project. While these

data would further strengthen the proposed HSP90–*atonal* pathway, we believe our new results provide a solid foundation for this model in *T. castaneum*.

4. Several conclusions overreach the presented data. Key findings (e.g., persistence of the trait, its selection, and its mechanistic basis) have low penetrance, limited replicates, and insufficient statistical support. Alternative hypotheses are not tested, and the environmental relevance of findings is unclear. Overall, stronger controls, clearer definitions, and more cautious interpretation are needed throughout. Statistical analyses are not robust. Effects on fitness are weak and may not be significant after multiple hypothesis testing.

response:

We believe that we could address these aspects with our revisions. Persistence of the *reduced-eye* trait was followed for many generations in our monomorphic lines, and stable phenotype frequencies were observed over many generations in our polymorphic lines, allowing us to use the F50 for Bulk Segregation Analysis. Even though effect sizes may appear to be weak, they are in an evolutionary relevant range, e.g. regarding fitness effects. Statistical analyses are not limited by insufficient replicate numbers, and multiple hypothesis testing was performed when needed.

Minor comments:

1. In place of “Hsp90-released trait” I suggest “Hsp90-buffered trait”; lines 38 and 74.

response:

Done – “HSP90-released trait” has been replaced with “HSP90-buffered trait” at the indicated lines. Now lines 40, 66 and 104

2. The keyword “plasticity” was not mentioned. Remove keyword or add explanation of the term in the text.

response:

Done – the keyword "plasticity" has been removed as requested.

3. Statistical significance, * should be replaced with increments **, ***, ****. A single asterisk (*) indicating a p-value less than 0.05 (significant), two asterisks (**) a p-value less than 0.01 (highly significant), and three asterisks (***) indicate a p-value less than 0.001 (very highly significant).

response:

Done – we have updated all significance indicators in the figures and legends to follow the standard format: $p < 0.05$ (*), $p < 0.01$ (**) and $p < 0.001$ (***), as requested.

4. Hsp90 is an epistatic regulator but no epistasis experiments were performed.

response:

Thank you for mentioning this aspect. While we did not perform classical genetic interaction (epistasis) assays in this study, our results are consistent with HSP90 acting as an epistatic

regulator. Specifically, knockdown of *Hsp83* released the *reduced-eye* phenotype; whole-genome sequencing and functional analysis identified the transcription factor ***atonal*** as the underlying gene; and direct knockdown of ***atonal*** robustly reproduced the same phenotype. Additionally, we observed a positive correlation between HSP90 and *atonal* expression, with *atonal* being downregulated after *Hsp83* knockdown. *Atonal* likely interacts epistatically with further, potentially also HSP90-affected proteins, but it is complex to address this in *T. castaneum* and beyond the scope of our study.

5. Abstract lines 33-34: “stably inherited without continued HSP90 disruption” implies all revealed variation was stable but the data show closer to 20% of the revealed variation is inherited and further declines in each generation so it is not that stable.

response:

We appreciate this comment and understand the potential confusion, which likely stems from the polymorphic lines that harbour the *reduced-eye* allele at a certain gene frequency (Fig. 2a, lower two panels). It is important to note the lines are polymorphic populations and not clonal lines. In these lines, the allele is stably inherited, but the phenotype frequency depends on the allele frequency in these populations. Importantly, it was possible to select for the *reduced-eye* phenotype and produce monomorphic lines that are likely fixed for the *reduced-eye* allele and thus show close to 100% *reduced-eye* phenotypes (Fig. 2a upper panel), which was maintained over all successive generations up to now. The original phrasing “stably inherited without continued HSP90 disruption” was used to convey that once established, the *reduced-eye* phenotype continued to persist in subsequent generations, even though HSP90 function was no longer perturbed. We have revised the abstract language to clarify the aspect of persistence in the population.

The revised abstract line now reads: line 35

“...we consistently revealed a reduced-eye phenotype that persisted in descendant lines across generations without continued HSP90 disruption.”

6. Figures should be revised to improve clarity and intuitiveness. In particular, the presentation of penetrance—such as in the HDAC inhibition example—is often unclear and could benefit from more straightforward visualization.

A more quantitative trait would be the number of ommatidia per animal, which although labor intensive seems to present with a great dynamic range and could foster a more robust statistical analysis approach. I understand that counting the number of ommatidia in each animal may not be feasible, so this is only a minor comment/suggestion.

response:

Thank you for the thoughtful suggestions regarding data visualization and quantification.

In response to the comment on HDAC inhibition (Extended Data Fig. 6):

We have revised the figure to improve its clarity and informativeness. Specifically, we now provide the **mean and standard error** for each group in addition to individual datapoints. Furthermore, we adjusted the **y-axis scale**, setting the lower bound to **50%** instead of 0%, to better focus on the region where all data points reside (60–100% penetrance). This zoomed-in view makes the small yet statistically significant effects more visually interpretable. The figure caption has also been updated to explicitly state this.

Regarding the suggestion to use ommatidia number as a quantitative trait:

We appreciate this idea and agree that ommatidial counts can, in many systems, provide a

detailed, quantitative metric. In our study, we performed a limited quantification of ommatidia number to support our conclusions (shown in Fig. 1e). Specifically, we manually counted the number of ommatidia in both compound eyes of adult beetles with clearly distinguishable eye morphology (n = 8) and used the average per beetle for analysis. The counts were compared between normal- and reduced-eye individuals using an **unpaired two-sample t-test**, which is regarded as a statistically robust method even with modest sample sizes.

However, we found that in *T. castaneum*, the number of ommatidia is relatively small and consistent between individuals of the same phenotype—approximately **40 ommatidia in normal eyes** and **~20 in reduced-eye phenotypes from ventral side**—resulting in **two discrete, stable categories**, rather than a broad dynamic range. In addition, reliable ommatidial quantification was only possible in adults with well-structured compound eyes. Many reduced-eye animals showed fused or misshapen ommatidia, making accurate counting unreliable in a substantial proportion of samples.

Due to these biological and technical constraints, we used **phenotypic penetrance** as our main metric, which was more scalable and practical for quantifying trait expression across all treatments and lines. Nonetheless, ommatidial counts in Fig. 1e provide important validation of the morphological distinctions between phenotypes.

We hope these revisions have addressed your concerns, and we are grateful for your comments that helped us strengthen the manuscript considerably.

Reviewer #3 (Remarks to the Author):

response:

We thank Reviewer #3 for his/her contribution to the peer review process and for participating in the Nature Communications initiative supporting Early Career Researchers. We appreciate the time and effort dedicated to co-reviewing our manuscript.

References

- Alford, Brian D., and Onn Brandman. 2018. 'Quantification of Hsp90 Availability Reveals Differential Coupling to the Heat Shock Response'. *Journal of Cell Biology* 217(11). doi:10.1083/jcb.201803127.
- Bailey, Timothy L., and Charles Elkan. 1994. 'Fitting a Mixture Model by Expectation Maximization to Discover Motifs in Biopolymers'. in *Proceedings of the 2nd International Conference on Intelligent Systems for Molecular Biology, ISMB 1994*.
- Borkovich, K. A., F. W. Farrelly, D. B. Finkelstein, J. Taulien, and S. Lindquist. 1989. 'Hsp82 Is an Essential Protein That Is Required in Higher Concentrations for Growth of Cells at Higher Temperatures.' *Molecular and Cellular Biology* 9(9):3919–30. doi:10.1128/MCB.9.9.3919.Updated.
- Brom, Krzysztof Roman, Bogdan Dolezych, Monika Tarnawska, Katarzyna Brzozowska, and Mirosław Nakonieczny. 2015. 'Expression of the Hsp40, Hsp70 and Hsp90 Proteins in Colorado Potato Beetle (*Leptinotarsa decemlineata* Say) after the Dimethoate Treatment'. *Journal of the Entomological Research Society* 17(2).
- Brown, Susan J., Teresa D. Shippy, Sherry Miller, Renata Bolognesi, Richard W. Beeman, Marcé D. Lorenzen, Gregor Bucher, Ernst a. Wimmer, and Martin Klingler. 2009. 'The Red Flour Beetle, *Tribolium Castaneum* (Coleoptera): A Model for Studies of Development and Pest Biology'. *Cold Spring Harbor Protocols* 4(8):1–12. doi:10.1101/pdb.emo126.
- Busto, M., B. Iyengar, and a R. Campos. 1999. 'Genetic Dissection of Behavior: Modulation of Locomotion by Light in the *Drosophila Melanogaster* Larva Requires Genetically Distinct Visual System Functions.' *The Journal of Neuroscience : The Official Journal of the Society for Neuroscience* 19(9):3337–44. doi:10.1523/JNEUROSCI.19-09-03337.1999.
- Chen, Bing, and Andreas Wagner. 2012. 'Hsp90 Is Important for Fecundity, Longevity, and Buffering of Cryptic Deleterious Variation in Wild Fly Populations'. *BMC Evolutionary Biology* 12(1):25. doi:10.1186/PREACCEPT-2105524459276497.
- Ding, Jian Hao, Lu Xin Zheng, Jie Chu, Xin Hao Liang, Jun Wang, Xiao Wen Gao, Fu An Wu, and Sheng Sheng. 2021. 'Characterization, and Functional Analysis of Hsp70 and Hsp90 Gene Families in *Glyphodes Pyloalis* Walker (Lepidoptera: Pyralidae)'. *Frontiers in Physiology* 12. doi:10.3389/fphys.2021.753914.
- Jeffery, William R. 2009. 'Regressive Evolution in *Astyanax* Cavefish'. *Annual Review of Genetics* 141(4):520–29. doi:10.1016/j.surg.2006.10.010.Use.
- Jez, J. M., J. C. Chen, G. Rastelli, R. M. Stroud, and D. V. Santi. 2003. 'Crystal Structure and Molecular Modeling of 17-DMAG in Complex with Human Hsp90'. *Chem Biol* 10(4):361–68. <http://www.ncbi.nlm.nih.gov/pubmed/12725864>.
- Kudryavtsev, Vladimir A., Anna V. Khokhlova, Vera A. Mosina, Elena I. Selivanova, and Alexander E. Kabakov. 2017. 'Induction of Hsp70 in Tumor Cells Treated with Inhibitors of the Hsp90 Activity: A Predictive Marker and Promising Target for Radiosensitization'. *PLoS ONE* 12(3):1–25. doi:10.1371/journal.pone.0173640.
- Lai, Yi Ting, Kevin D. Deem, Ferran Borràs-Castells, Nagraj Sambrani, Heike Rudolf, Kushal Suryamohan, Ezzat El-Sherif, Marc S. Halfon, Daniel J. McKay, and Yoshinori Tomoyasu. 2018. 'Enhancer Identification and Activity Evaluation in the Red Flour Beetle, *Tribolium Castaneum*'. *Development (Cambridge)* 145(7). doi:10.1242/dev.160663.
- Laughlin, S. B., R. R. de Ruyter Van Steveninck, and J. C. Anderson. 1998. 'The Metabolic Cost of Neural Information'. *Nature Neuroscience* 1(1):36–41. doi:10.1038/236.
- Liang, Chen, Lifang Li, Hang Zhao, Mingxian Lan, Yongyu Tang, Man Zhang, Deqiang Qin, Guoxing Wu, and Xi Gao. 2023. 'Identification and Expression Analysis of Heat Shock Protein Family Genes of Gall Fly (*Procecidochares Utilis*) under Temperature Stress'. *Cell Stress and Chaperones* 28(3). doi:10.1007/s12192-023-01338-9.

- Mau, Christine, Heike Rudolf, Frederic Strobl, Benjamin Schmid, Timo Regensburger, Ralf Palmisano, Ernst H. K. Stelzer, Leila Taher, and Ezzat El-Sherif. 2023. 'How Enhancers Regulate Wavelike Gene Expression Patterns'. *ELife* 12. doi:10.7554/eLife.84969.
- Moran, Damian, Rowan Softley, and Eric J. Warrant. 2015. 'The Energetic Cost of Vision and the Evolution of Eyeless Mexican Cavefish'. *Science Advances* 1(8):e1500363–e1500363. doi:10.1126/sciadv.1500363.
- Niven, Jeremy E. 2015. 'Neural Evolution: Costing the Benefits of Eye Loss'. *Current Biology* 25(19):R840–41. doi:10.1016/j.cub.2015.08.050.
- Okawa, Haruhisa, Alapakkam P. Sampath, Simon B. Laughlin, and Gordon L. Fain. 2008. 'ATP Consumption by Mammalian Rod Photoreceptors in Darkness and in Light'. *Current Biology* 18(24):1917–21. doi:10.1016/j.cub.2008.10.029.
- Park, T. 1934. 'Observations on the General Biology of the Flour Beetle, *Tribolium Confusum*'. *Quarterly Review of Biology* 9:36–54.
- Peuss, R., Hendrik Eggert, Sophie A. O. Armitage, and Joachim Kurtz. 2015. 'Downregulation of the Evolutionary Capacitor Hsp90 Is Mediated by Social Cues'. *Proceedings of the Royal Society B: Biological Sciences* 282(1819):20152041–20152041. doi:10.1098/rspb.2015.2041.
- Pointer, Michael D., Matthew J. G. Gage, and Lewis G. Spurgin. 2021. 'Tribolium Beetles as a Model System in Evolution and Ecology'. *Heredity* 126(6):869–83. doi:10.1038/s41437-021-00420-1.
- Ri as-Sánchez, David F., Jake Morris, Camilo Salazar, Carolina P. ar do-Díaz, Richar M Merrill, Stephen H. Montgomery, and Richard M Merrill. 2025. 'Repea Ted e Volution of Reduced Visual in Vestment a t the Onset of Ecological Speciation in High-Altitude Heliconius Butterflies'. *Evolution Letters* 1–9. doi:10.1093/e.
- Rutherford, S. L., and S. Lindquist. 1998. 'Hsp90 as a Capacitor for Morphological Evolution'. *Nature* 396(6709):336–42. doi:10.1038/24550.
- Trepel, Jane, Mehdi Mollapour, Giuseppe Giaccone, and Len Neckers. 2010. 'Targeting the Dynamic HSP90 Complex in Cancer'. *Nature Reviews Cancer* 10(8):537–49. doi:10.1038/nrc2887.
- Waddington, C. H. 1942. 'Canalization of Development and the Inheritance of Acquired Characters'. *Nature* 150(3811):563–65. doi:10.1038/150563a0.
- Zhou, Dan, Yuan Liu, Josephine Ye, Weiwen Ying, Luisa Shin Ogawa, Takayo Inoue, Noriaki Tatsuta, Yumiko Wada, Keizo Koya, Qin Huang, Richard C. Bates, and Andrew J. Sonderfan. 2013. 'A Rat Retinal Damage Model Predicts for Potential Clinical Visual Disturbances Induced by Hsp90 Inhibitors'. *Toxicology and Applied Pharmacology* 273(2):401–9. doi:10.1016/j.taap.2013.09.018.

RESPONSE TO THE REVIEWER COMMENTS _2nd Round

Dear Reviewers,

We sincerely thank you for your continued evaluation of our manuscript entitled “*HSP90 as an Evolutionary Capacitor Drives Adaptive Eye Size Reduction via Atonal.*” We appreciate the additional feedback provided and have addressed the remaining points to further improve the clarity and robustness of our work.

Reviewer #1 (Remarks to the Author):

My biggest concerns in the previous version of the manuscript were the links between Hsp90 levels, phenotypic variation, and environmental conditions. In their response letter, the authors do a convincing job of explaining that the conditions that affected Hsp90 availability need not be the same conditions that select upon phenotypes and genotypes revealed by Hsp90 depletion. Further, they explained the link between measuring expression level of Hsp68 instead of Hsp83. By and large, I am satisfied with their responses to my initial comments and suggestions, and I am pleased by their inclusion of additional data and consideration of my comments. I have only a few suggestions for further clarifying these points in the text.

response:

We are grateful for the reviewer’s positive evaluation and are pleased that our previous revisions addressed the major concerns. As suggested, we have incorporated additional clarifications in the relevant sections to further strengthen the connections between HSP90 availability, phenotypic variation, and environmental context.

L87: Consider mentioning here that selection could be acting in the Hsp90 affecting environment or a different environment. “...differences on which selection can act either in the disrupting environmental conditions or other conditions altogether.”

response:

The sentence has been revised as follows (L51 with “No Markup” as the adjusted option): “...differences on which selection can act **either in the disrupting environmental conditions or other conditions altogether.**”

L186 consider adding a reminder here that continuous light COULD be encountered by these beetles in their human-commensal environment.

response:

We added the following (L160 with “No Markup” option): “...continuous light stress, a situation that could be encountered by these beetles in their human-commensal environment.”

L208 Consider changing “certain” to “some, albeit unknown, frequency”

response:

This change has been made (L180 with “No Markup” option).

L321 “...unmasking cryptic genetic variation and producing new phenotypes.” To be explicit about the consequence of that CGV being uncovered.

response:

We added “and producing new phenotypes” (L293 with “No Markup” option).

Reviewer #2 (Remarks to the Author):

All my comments have been fully addressed. I congratulate the authors on their beautiful work and approve of its publication.

response:

We sincerely thank the reviewer for their very helpful previous comments that substantially improved our manuscript and are grateful for this positive feedback.